# 🍾 $\mathcal{C}ola$: A Benchmark for Compositional Text-to-image Retrieval

**Arijit Ray**[1,2]     Filip Radenovic[2]     Abhimanyu Dubey[2]     Bryan A. Plummer[1]
Ranjay Krishna[2,3]          Kate Saenko[1,2]

{array, bplum, saenko}@bu.edu,{filipradenovic, dubeya}@fb.com,
ranjay@cs.washington.edu

[1]Boston University, [2]Meta AI (FAIR), [3]University of Washington

## Abstract

Compositional reasoning is a hallmark of human visual intelligence. Yet despite the size of large vision-language models, they struggle to represent simple compositions by combining objects with their attributes. To measure this lack of compositional capability, we design $\mathcal{C}ola$, a text-to-image retrieval benchmark to **C**ompose **O**bjects **L**ocalized with **A**ttributes. To solve $\mathcal{C}ola$, a model must retrieve images with the correct configuration of attributes and objects, and avoid choosing a distractor image with the same objects and attributes but in the wrong configuration. $\mathcal{C}ola$ contains about 1.2k composed queries of 168 objects and 197 attributes on around 30K images. Our human evaluation finds that $\mathcal{C}ola$ is 83.33% accurate, similar to contemporary compositionality benchmarks. Using $\mathcal{C}ola$ as a testbed, we explore empirical modeling designs to adapt pre-trained vision-language models to reason compositionally. We explore 6 adaptation strategies on 2 seminal vision-language models, using compositionality-centric test benchmarks - $\mathcal{C}ola$ and CREPE. We find the optimal adaptation strategy is to train a multi-modal attention layer that jointly attends over the frozen pre-trained image and language features. Surprisingly, training multimodal layers on CLIP performs better than tuning a larger FLAVA model with already pre-trained multimodal layers. Furthermore, our adaptation strategy improves CLIP and FLAVA to comparable levels, suggesting that training multimodal layers using contrastive attribute-object data is key, as opposed to using them pre-trained. Lastly, we show that $\mathcal{C}ola$ is harder than a closely-related contemporary benchmark, CREPE, since simpler fine-tuning strategies without multimodal layers suffice on CREPE, but not on $\mathcal{C}ola$. However, we still see a significant gap between our best adaptation and human accuracy, suggesting considerable room for further research. Project page: https://cs-people.bu.edu/array/research/cola/

## 1   Introduction

Compositionality is a fundamental characteristic of human intelligence, allowing us to elicit "the meaning of the whole [as] a function of the meanings of its parts" [9]. In language, the whole is a sentence made up of words like nouns and adjectives. In vision, the whole is an image made up of visual elements like objects and attributes [27, 19]. For example, the expression "round white table" is a composition of the noun "table" and adjectives "round" and "white", visually represented in the leftmost photo of Fig. 1. Recent work has consistently identified that this type of compositionality—that between objects and their attributes—is something existing vision-language models struggle to represent [54, 34, 24]. Instead, they disperse attributes and ground them to distractor objects; for instance, they incorrectly match "round white table" to the second left photo in

37th Conference on Neural Information Processing Systems (NeurIPS 2023) Track on Datasets and Benchmarks.

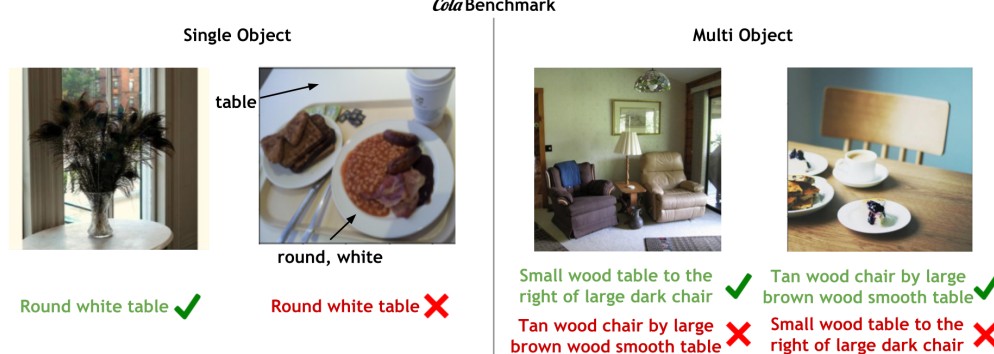

Figure 1: We present ⚗ $\mathcal{C}ola$, where a model has to Compose Objects Localized with Attributes. To solve $\mathcal{C}ola$, a model must match the correct image to the correct caption, not a distractor image with the same objects and attributes but in the wrong configuration. We explore the design space of possible mechanisms to adapt existing models to this task; we show that a simple multimodal adaptation method to finetune pre-trained vision-language representations works best.

Fig. 1 by grounding the attribute "round" to the round plate instead of the table. Queries involving two objects are even more challenging, see Fig. 1 (right).

In this paper, we study the ability of large vision-language models to **C**ompose **O**bjects **L**ocalized with **A**ttributes ($\mathcal{C}ola$). Unlike related baselines that study compositionality using relationships [54] and scene graphs [34], we focus on attribute-object bindings, since finding objects with correct attributes is crucial in many applications. For example, an embodied AI assistant told to clean the "small wood table to the right of the large dark chair" should *not* start cleaning the "tan wood chair by the large brown wood smooth table." Additionally, object-attribute bindings should fundamentally be easier than compositions with relationships or scene graphs. However, we find that existing models still struggle with this simpler binding.

To explore these issues, we propose the $\mathcal{C}ola$ benchmark for composing objects localized with multiple attributes. $\mathcal{C}ola$ contains two kinds of compositions: single-object queries (Fig. 1 left) and multi-object (Fig. 1 right). In each case, a model should associate the objects in the query with the correct attributes and ignore difficult distractor compositions where the query attributes are attached to distractor objects. Multi-object queries are harder since they showcase more compositions.

Unlike an image-to-text setting in a contemporary benchmark CREPE [34], $\mathcal{C}ola$ evaluates models using text-to-image, where text queries are used to retrieve the correct image from a set of images. This is consistent with previous benchmarks for vision-language models [20, 41, 62]. Further, text-to-image retrieval is harder than image-to-text retrieval because image encoders are weaker at distinguishing fine-grained differences in images for a given text than text encoders are at distinguishing fine-grained text [54]. Moreover, text-to-image is better aligned with practical applications, such as a user giving text instructions to a machine to find certain objects.

Using $\mathcal{C}ola$ as a development testbed, our experiments add to the ongoing discussion that pre-trained vision-language models perform poorly on compositions [54]. Hence, we explore 6 finetuning strategies on 2 seminal vision-language models- CLIP [42] and FLAVA [51]- to find adaptation strategies that encourage compositionality the most. We finetune using 3 datasets (GQA [17], CLEVR [23], and PACO [43]) and evaluate on 2 testbeds ($\mathcal{C}ola$ as well as CREPE [34]). While exploring effective pre-training strategies is another valid avenue of exploration, we limit our work to adaptation strategies since training a model from scratch is expensive and can only be executed by a handful of research organizations.

We find that the best-performing architectural choice during adaptation to be a multi-modal transformer encoder-decoder [52, 51, 25] to further encode the visual and language representations from the pre-trained model. Multimodal adaption performs significantly better than tuning the unimodal encoders (that encode just vision or language features), or tuning the prompt. Surprisingly, this adaptation improves both CLIP and FLAVA to produce comparable finetuned models, even though our CLIP model has fewer parameters and FLAVA was already pre-trained using multi-modal transformer

layers. This suggests that training multimodal layers using contrastive attribute-object data is key, as opposed to using them pre-trained on web data. Our adaptation also significantly outperforms standard ways to adapt/tune foundational models such as prompt-tuning [63], linear-probing [2], or tuning a comparable number of split-encoder layers. Similar to recent work identifying that structural compositionality is present in models but absent in their representations [30], our work finds that while pre-trained representations might not exhibit compositionality, they can be adapted to do so. However, the stark difference between human accuracy and our best adaptation suggests considerable room for further research using our benchmark.

## 2   Related Work

**Compositionality and image retrieval.** Compositionality is a key aspect of human intelligence [10], especially in vision and language [7]. Vision-language compositionality has been explored for visual question answering [1], composed image retrieval (*e.g.*, X in the style of Y) [48], and generation [55]. Compositionality is one crucial aspect of improving robustness to diverse queries, a theme heavily explored in the vision-language community [45, 44, 50, 1, 8]. With the recent popularity of foundation models, various works focus on testing their compositional reasoning [34, 54, 24, 60]. Compared to CREPE [34] and ARO [60], a model must distinguish between difficult images in our case. Text-to-difficult-images is harder because distinguishing between difficult images (for a given caption) is harder than distinguishing between difficult captions [54]. Whereas benchmarks like Winoground [54] primarily evaluate broad and complex relational compositionality (*e.g.*, "man hugs woman from behind" vs "woman hugs man from behind"), we specifically focus on attribute object bindings in queries. This is motivated by practical applications such as an embodied agent trying to retrieve a custom object (like "a metal wrench with a red rubber handle") in a cluttered workspace with similar distractor objects [38]. Most works in the area of attribute-object image retrieval either focus on single attributes [18, 40] or multiple attributes in very niche domains with centered images and plain backgrounds of dresses [15], animals [58], shoes[59], or birds [56]. In contrast, we focus on scenes with multiple objects and attributes where distractor objects also have the same attributes.

**Vision-language aligment.** Recently, there has been a flurry of image-text alignment models to learn the similarity of matched images and text in various ways. Some models use separate unimodal encoders [42, 21] for the image and text, whereas some [51, 32, 3, 11, 49] use multimodal encoders as well. Various strategies such as hard negative mining [32], concept distillation [39], and maintaining the momentum of image-text mappings [16] have been employed to push performance. We focus on testing and improving the attribute-object binding capability of such models and choose the most seminal model, CLIP [42], which is widely adopted in various concurrent vision-language research/applications [53, 26, 46]. Our approaches do not use any box annotations unlike recent text localization models [25, 33, 61], which we also see to underperform on text-to-image retrieval.

**Adapting foundational models.** Since training a new VLM from scratch is expensive, we wish to formulate a simple adapter that improves the compositional attribute-object binding. Various works explore adapting foundation models [4] with prompt-tuning [63], linear-probing [2], and fine-tuning with residual connections [13]. Prompt-tuning [31] learns the embedding layer of the word inputs and keeps the model frozen. Inspired by the success of prompt-tuning [31], some works have also explored prompting in the vision [22, 5] and vision-language [63, 47], and also for single attribute-object compositions [36]. Our optimal finetuning strategy improves significantly over prompt and fine-tuning for attribute-object compositions in even more difficult settings. Our multi-modal strategies are similar to MAPL [35], except our lightweight adapter attends over language and vision representations, whereas MAPL only attends over language.

## 3   🔒𝒞𝑜𝑙𝑎 benchmark

Our goal is to adapt vision-language features to improve the compositional binding of attributes to objects. Specifically, we aim to improve the classification of a query involving single or multiple objects with multiple attributes in an image. Images and language are composed of atomic concepts such as attributes and objects. The atomic concepts ("square", "plate") form certain compounds ("square plate"), and then the scene is a combination of various such compounds ("square plate on white table"). Hence, we create a benchmark where we form queries using compositions of such

atoms and test a model's ability to distinguish between images that correctly contain the atoms in the *correct* composition to distractor images that contain them in the *wrong* composition. In total, the $\mathcal{C}ola$ benchmark contains about 1236 composed queries from 168 objects and 197 attributes on around 30K images from 4 datasets.

$\mathcal{C}ola$ contains two query types discussed below: single-object compounds and multi-object queries.

**Retrieval using single-object queries.** Single-object queries have multiple attributes grounded on one object. For example, the query "square white plate," which is of the form, $Q = a_1 a_2 o$, where $a_i \in A$ is drawn from a finite set of possible attributes and $o \in O$ is similarly a category drawn from a finite set of objects. With this query, a model should associate the images with the correct attachment of attributes ("square," "white") to the object ("plate"), and ignore incorrect attachments of the same attributes and objects (like square table but not square plates). Hence, the task is a text query for image retrieval among difficult distractors. We first create a list of queries with more than one attribute for an object. Next, we curate a set of images where at least one of the query words is present in the image. For example, for "square white plate," all images containing "square" objects, "white" objects, or "plates" are in the list of images to retrieve from. The goal of the retrieval problem is to score the images having the correct attachment of the attributes to the query higher than others. We build the test set for single object queries using three datasets with object and attribute annotations: 1) GQA [17]: After filtering for objects with at least 1 attribute annotated, we have 320 single-object queries composed of 114 objects and 114 attributes on 1952 images. The objects and attributes comprise common objects, making this split useful for practical applications. 2) CLEVR [23]: We have 3 object shapes - cubes, cylinders and spheres, composed with 8 colors, 2 materials, and 2 sizes on 15K images. 3) PACO [43]: This split consists of objects similar to GQA. We have 400 queries composed of 51 objects and 61 attributes on 7921 images.

**Retrieval with multi-object queries.** Drawing on existing literature [34], a multi-object query contains multiple objects, each with its own set of attributes. For example, "square white plate on top of brown wooden table," which is of the form, $Q = a_1 a_2 o_1 + a_3 a_4 o_2$, where $a_i \in A$ is drawn from a finite set of possible attributes and $o_j \in O$ from a finite set of objects. In this setting, we want to check if the model gets confused with the wrong configuration of objects and attributes. Thus, we find distractor image-query pairs where the attributes and objects are switched. An example image for a query $Q = a_1 o_1 + a_2 o_2$ would be of the form $I' = a_2 o_1 + a_1 o_2$. In other words, we switch the attributes of the two objects. We curate these distractors to ensure that $o_1 \neq o_2$ and $a_1 \neq a_2$. The retrieval task, framed with this formalism, is to rank the correct images for the correct captions such that it is ranked higher than the distractor images: to learn a relevance encoding $f(I, Q)$ for image $I$ and query $Q$ such that $f(I, Q) > f(I', Q)$ & $f(I', Q') > f(I, Q')$. The test set is built using test split of the Visual Genome [28] dataset.

**Filtering $\mathcal{C}ola$ multi-object with crowd workers.** We use the object, attribute, and relationship annotations in the Visual Genome dataset [28] to create the multi-object queries. We filter the image-caption pairs with object and attribute compositions swapped as described above. We conduct a human-annotated cleaning of this filtered test set. We display the images $I$ and $I'$ and queries $Q$ and $Q'$ to 10 crowd workers and ask them to choose which image is most relevant to which query. We only keep the image-query pairs where the majority of crowd workers can correctly assign the correct image to the query. After filtering, we are left with 210 data points (840 image-query pairs) with 1680 possible image-query matches. The human agreement (accuracy) on our validation set is 83.88% - an average of 8.33 out of 10 humans agree that the first image matches to the first caption and second image to the second caption. Some qualitative examples are provided in Fig. 2a.

## 4 Exploring finetuning strategies with $\mathcal{C}ola$

Given an image ($I$) and a query ($Q$), $\mathcal{C}ola$ evaluates a model $f(I, Q)$ by measuring how well it associates the correct image to the input query. Existing pre-trained models don't perform well on this task since they fail to distinguish fine-grained differences in attribute-object compositions. Hence, we explore finetuning strategies that use a dataset of image-language pairs where the language descriptions contain objects and attributes. Details of finetuning datasets are described in Sec. 5.3. We follow the standard finetuning paradigm by sampling batches of images and text from the attribute-object $\mathcal{C}ola$ finetuning dataset. Specifically, we match the correct images to the correct queries in each batch and minimize the NCELoss typically used in contrastive learning [42, 51]. This finetuning

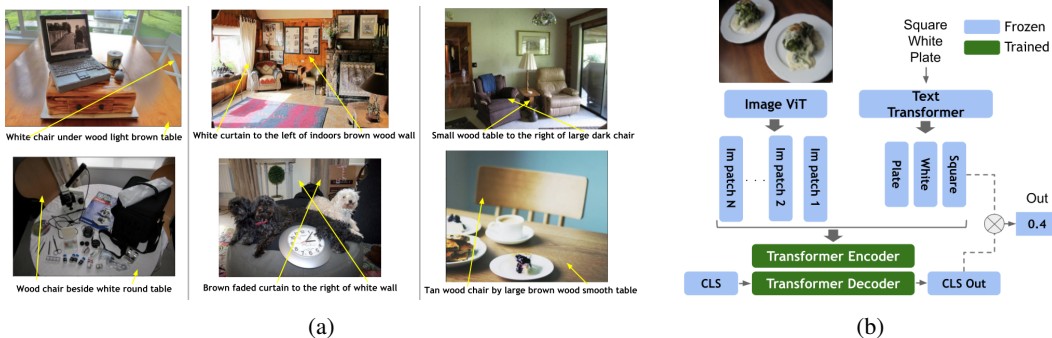

(a)             (b)

Figure 2: **a)** $\mathcal{C}ola$ **multi-object setting validation set:** a human-cleaned difficult validation set for testing attribute-object binding. The two images have similar objects and attributes but in different configurations. A model must match the correct images to the correct captions. **b) The optimal adaptation strategy (MM-Adapter):** a lightweight multimodal transformer encoder on top of frozen pre-trained encoders. The multimodal encoder crafts a stronger representation by cross-attending to image patches and text tokens to attach the correct attributes to the correct objects. The stronger representation is then trained to align with the frozen text representation.

step aims to improve the compositional binding of attributes and objects in pretrained vision-language features. This is in contrast to training the multimodal layers on random batches of web image-text data.

**Disjoint finetuning strategies** Since CLIP [42] is commonly used for various tasks and is one of the more lightweight vision-language foundation models, we focus most of our finetuning strategies with CLIP in mind. Although, we also later use these finetuning strategies on the newer and larger FLAVA [51] model. CLIP [42] consists of two encoders: one that encodes the input image and one that similarly encodes the input text. The output representations of the two modalities are used as separate embeddings for the image and text. To adapt these models for a specific task/capability, researchers commonly use linear-probing or prompt-tuning. Linear-probing trains linear layers on top of the frozen visual and text encoders using the finetuning dataset. Prompt-tuning learns the word embeddings of the query to adapt to the finetuning dataset domain without changing the weights of the model [63]. Other methods fine-tune the later layers of both the encoders. All these adaptation methods tune the parameters of the two encoders separately.

**Joint multimodal strategies** We hypothesize that the above common adaptation strategies don't appropriately capture the cross-modal interaction required for strong attribute-object binding. However, CLIP is significantly more lightweight than recent multimodal models. Hence, we explore lightweight multi-modal adaptation strategies to adapt CLIP. We describe the best-performing multimodal adaptation strategy found in our experiments which is also depicted in Fig. 2b. This multi-modal adaptation borrows a transformer encoder-decoder [52, 51, 25] to attend over the image and language representations jointly. Let $M = [I; Q]$, denote the concatenated image patch representations extracted from CLIP's visual encoder and the token-level representations from the query. We compute a self-attention over $M$ using a transformer encoder $A = Att(M)$. Finally, we use a classify token (a token randomly initialized), referred to as `[CLS]`, that cross-attends [1] to all the self-attended features $A$ using a transformer decoder to produce $out_{MM}$. This type of cross-attending to self-attended features is similar to FLAVA[51]/MDETR [25].

The standard practice in most multi-modal encoder-based prediction models would be to learn a linear layer to classify the output `[CLS]` token embedding [25, 51, 32, 11]. However, instead of learning a linear predictor, we compute the cosine similarity of `[CLS]` to the representations of the query tokens: $q_i$ from the frozen text encoder, $Q$. We posit that aligning to a frozen text encoder trained on larger scale data will act as a regularizer, helping performance on unseen compositions. Hence, the final score for a given image-query pair is $f(I, Q) = \frac{1}{N_q} \sum_i^{N_q} out_{MM} \odot q_i$ where $N_q$ is the number of tokens in the query. These two ablations are referred to as **MM-pred** and **MM-adapter**

---

[1]query comes from `[CLS]`, and the keys and values come from self-attended features, $A$.

("adapter" since we can think of the latter as adapting the image features to align better with text) in our experiments.

We also tried various flavors of computing a multi-modal adaptation inspired by FIBER [11] and ALBEF [32], which use cross attention between text and image. We would like to stress that while exploring newer ways to fuse vision and language features is a valid avenue of research, we are interested in exploring the common themes in current fusion methods that encourage compositionality the most to drive future research. We report the best method for simplicity and include the accuracies from other strategies in Table 3 in the supplemental.

## 5 Evaluation Setup

We evaluate models on the two types of queries described in Sec. 3 on $\mathcal{Cola}$ and CREPE [34] datasets. All models are trained using Pytorch [37] and use the Huggingface transformers library [57]. Implementation details such as hyperparameters are provided in Sec. 3 of the supplementary.

### 5.1 Metrics

**Single object queries.** For this type of query, we report the $\mathcal{Cola}$ MAP[2], the mean average retrieval precision over difficult distractors. We further differentiate the mAP between **seen** and **unseen** queries. We split $\mathcal{Cola}$ into seen and unseen sets by removing some attribute-object pairs from the training set. For example, "square white plate" is unseen if this combination is absent in finetuning; however, "square white bowl" or a "square plate" may be present. In our test set, we have 320 (150 unseen, 170 seen) queries on 1950 images for GQA, 96 (32 unseen, 64 seen)[3] queries on 22500 images for CLEVR, and 400 (200 seen, 200 unseen) queries on 7921 images in PACO.

**Multi-object queries.** Recall that we have two images and two captions, and the task is to match the correct caption to the correct image. If we denote the prediction score for an image and query to be $f(I, M)$, we regard a prediction to be correct if $f(I, M) > f(I', M)$ & $f(I', M') > f(I, M')$, where images $I$ and $I'$ are paired to captions $M$ and $M'$ respectively. Using this criterion, we compute the $\mathcal{Cola}$ multi-object accuracy. The random accuracy is $25\%$ since there are four ways to match the two captions to the two images. We also evaluate on a contemporary dataset, CREPE [34], where the task is inverse. For CREPE, we compute an image-to-text (I2T) accuracy, where a model must match the correct text from two choices to the given image. Note that there is only one image for the two caption choices in CREPE [34]. The random accuracy is $50\%$ since it is a binary task.

### 5.2 Explored finetuning strategies

Recall that the best-performing finetuning approach we found is a multimodal adaptation (**MM-Adapter**) for tuning pre-trained image and text features as described in Sec. 4. We compare against popular tuning methods like linear probing, tuning the prompt embeddings (prompt-tuning), and fine-tuning the whole model (FT all) or the last two layers (FT Late). These adaptations are applied separately to the base model for comparison. More details are in the supplementary (Sec. 3). Since our adaptation uses multimodal attention, we also compare it to a seminal model that uses multimodal attention in pretraining. We chose FLAVA [51] since it is one of the recent models after CLIP which is bigger, more accurate [51] and has easily available pre-trained weights.

### 5.3 Finetuning datasets

The $\mathcal{Cola}$ training sets are also built in the same way as described in Sec. 3 using the training splits of GQA [17, 28], CLEVR [23] PACO [43], and Visual Genome [28]. For GQA, the training split contains 1381 objects and 601 attributes that compose 27078 queries on 74K images. For CLEVR, we have 3 shapes composed with 8 colors and 2 sizes on 70K images. Finally, for PACO, we have 75 objects and 55 attributes that compose 18696 queries on 37883 images. The $\mathcal{Cola}$ multi-object

---

[2]we also computed the F1 score and see trends remain the same; more details in the supplementary, Sec. 2, page 4.

[3]we report "seen" only on 32 classes to avoid disbalance with unseen. "All" MAP is on all 96 classes. "Seen" trends hold same with 64 classes as well; more details in appendix.

| | GQA | | | CLEVR | | | PACO | | |
|---|---|---|---|---|---|---|---|---|---|
| | **All** | **Unseen** | **Seen** | **All** | **Unseen** | **Seen** | **All** | **Unseen** | **Seen** |
| a. CLIP | 36.53 | 39.06 | 34.24 | 15.38 | 15.01 | 15.32 | 12.21 | 8.64 | 15.79 |
| b. +Linear | 40.44 | 42.87 | 38.24 | 47.96 | 29.43 | 46.75 | 14.22 | 6.75 | 21.68 |
| c. +prompt-tune | 37.40 | 40.69 | 34.43 | 29.61 | 23.17 | 28.05 | 12.76 | 5.92 | 19.61 |
| d. +FT all | 38.81 | 40.85 | 36.95 | 52.32 | 19.00 | 47.95 | 14.58 | 6.49 | 22.66 |
| e. +FT late | 42.19 | 44.61 | 40.01 | 64.06 | 27.53 | 67.48 | 15.66 | 8.74 | 22.58 |
| f. **+MM-Pred (us)** | **45.99** | **48.6** | **43.64** | **75.80** | **51.98** | **80.72** | **15.49** | **8.00** | **22.94** |
| g. **+MM-Adapter (us)** | **46.83** | **48.86** | **44.99** | **88.21** | **89.52** | **77.00** | **18.56** | **11.47** | **25.66** |
| h. FLAVA | 39.65 | 42.18 | 37.37 | 15.41 | 13.27 | 15.93 | 12.53 | 7.29 | 17.76 |
| i. +Linear | 37.07 | 39.96 | 34.46 | 19.30 | 17.53 | 18.52 | 11.65 | 7.90 | 15.39 |
| j. +FT-late | 39.58 | 42.26 | 37.16 | 77.95 | 72.72 | 66.42 | 12.82 | 5.79 | 19.84 |
| k. **+MM-Pred (us)** | **47.12** | **51.53** | **43.13** | **90.43** | **85.78** | **86.07** | **18.57** | **10.71** | **26.44** |
| l. **+MM-Adapter (us)** | **48.54** | **52.55** | **44.91** | **91.10** | **86.64** | **87.39** | **19.36** | **11.16** | **27.55** |

Table 1: mAP results on the $\mathcal{C}ola$ single object compounds setting. Our multimodal adaptation (MM-Adapter) performs better than common tuning methods. Further, multimodal attention to adapt the image representation (MM-Adapter) generalizes better than using it simply as a prediction head (MM-Pred). MM-Adapter on CLIP is better than tuning the pre-trained multimodal attention layers of the bigger FLAVA (+FT late). MM-Adapter further improves FLAVA.

compounds training split has 551,980 multi-object compounds on 71,174 images. Only the test split is cleaned using human annotations. For datasets built on GQA [17] and Visual Genome [28], we leverage the annotations to explore the effects of different kinds of data queries. We use the region descriptions (denoted as **RegionCap** in the tables) in Visual Genome to test if linguistic diversity helps over templated captions ($\mathcal{C}ola$ single objects and multi-object). We also compare to hard negatives from the $\mathcal{C}ola$ multi-object pairs. We finally have a combined setting where we use all data.

## 6 Results

Recall that we evaluate on two settings for $\mathcal{C}ola$ - the single-object compounds setting and the multi-object compounds setting as defined in Sec. 3. We discuss the quantitative results below and some qualitative results are shown in Fig. 3.

### 6.1 $\mathcal{C}ola$ **Single-object retrieval**

**Multimodal adaptation is more effective than other tuning methods:** In Table 1, compared to prompt-tuning (row c), fine-tuning all of CLIP (row d), or fine-tuning a few of the later layers (row e), tuning a multimodal attention layer of same/lesser parameters has higher mAP (row f and g). Linear probing (row b), although cheaper, significantly underperforms. This is not surprising since multimodal attention over the image regions and text tokens offers more flexibility to the model to learn to bind the right attributes to the right object region. Tuning the whole model is also worse than tuning the later unimodal layers (row d vs e). This might be because fine-tuning the whole model requires larger batch sizes with significantly more data. In Fig 3, we show the comparison of tuning unimodal layers vs our multimodal adaptation (since tuning the unimodal layers is closest in performance). Qualitative examples from each of the other adaptation methods are displayed in Figs. 5-11 in the supplementary.

**Using pre-trained multimodal attention layers is not enough - training them on attribute-object compositions is key:** In Table 1, we see that MM-Adapter (row g) on CLIP ViT B-32 (151 M params) outperforms tuning the last two multimodal layers of FLAVA B-16 [51] (241M params) model (row j) or tuning a linear probe (row i). Surprisingly, tuning the last two FLAVA multimodal layers (row j) is worse than replacing them and training using MM-Pred and MM-Adapter layers (rows k and l). This suggests that training multimodal layers during adaptation (as opposed to pre-training) is key.

**MM-Adapter is better than using multimodal attention as a prediction head:** Recall that one of the ablations of our approach is aligning the output of the multimodal encoder to the frozen

| | Multi-Obj Acc ↑ | |
|---|---|---|
| | **T2I** $\mathcal{C}ola$ | **I2T** **CREPE** [34] |
| - Random | 25.00 | 50.00 |
| - Human | 83.88 | - |
| o. CLIP | 21.42 | 77.43 |
| a. + Linear | 30.47 | 87.35 |
| b. + Prompt-tune | 27.14 | 80.81 |
| c. + FT all | 34.76 | 82.39 |
| d. + FT late | 36.19 | 87.14 |
| e. + MM-Pred (our) | 41.42 | 77.84 |
| f. + MM-Adapter (our) | 40.95 | 87.02 |
| g. FLAVA | 24.76 | 65.10 |
| h. + Linear | 22.38 | 55.10 |
| i. + FT late | 22.38 | 58.11 |
| j. + MM-Pred (our) | 39.04 | 81.37 |
| k. + MM-Adapter (our) | 40.47 | 74.81 |

(a)

| Data Type | Single-Object $\mathcal{C}ola$ GQA | | |
|---|---|---|---|
| | **All** | **Unseen** | **Seen** |
| a. RegionCap | 0.4711 | 0.4965 | 0.4481 |
| b. SingleObj | 0.4683 | 0.4886 | 0.4499 |
| c. + MultiObj | 0.4641 | 0.4795 | 0.4501 |
| d. + HardNeg | 0.4688 | 0.4843 | 0.4548 |
| e. Combined | **0.4788** | **0.4983** | **0.4612** |
| | **Multi-Object** | | |
| | $\mathcal{C}ola$ | **CREPE** | |
| a. RegionCap | 0.3114 | 0.8833 | |
| b. SingleObj | 0.2745 | **0.9023** | |
| c. + MultiObj | **0.3975** | 0.8702 | |
| d. + HardNeg | 0.3483 | 0.8775 | |
| e. Combined | 0.3893 | 0.8798 | |

(b)

Table 2: a. Results on our multi-object compounds setting on our $\mathcal{C}ola$ task and CREPE. Simpler methods suffice on CREPE, but not on $\mathcal{C}ola$, suggesting that $\mathcal{C}ola$ is harder. Red-orange-yellow is in decreasing accuracy order. MM-Adapter and MM-Pred on CLIP perform well on average on both. b. Table showing the effect of the data type used in the contrastive batch training. Having multi-object captions in the data helps $\mathcal{C}ola$ performance while maintaining CREPE performance.

text embedding, making it a multimodal image-feature "adapter" (MM-Adapter). This contrasts to using the multimodal module as a prediction head with a linear layer (MM-Pred). As shown in Table 1, MM-Adapter outperforms MM-Pred (row g vs f), especially on unseen classes and on exhaustively annotated datasets like CLEVR [23] and PACO [43]. We posit that aligning to the frozen text representation acts like a regularizer since it was pre-trained on more data.

## 6.2 Multi-object retrieval

**Simpler methods suffice for CREPE, but not for $\mathcal{C}ola$, suggesting that $\mathcal{C}ola$ is a harder task:** As shown in Table 2a, linear probing (row a) or simple fine-tuning (rows c and d) suffice for CREPE [34]. However, our MM-Adapter improves further on $\mathcal{C}ola$ (row f vs row a, b, c, d), while maintaining performance on CREPE. This also suggests that text-to-image matching is harder than image-to-text matching, which is also reflected in Winoground [54].

**Baseline CLIP and FLAVA perform below chance:** If we evaluate off-the-shelf CLIP [42] and FLAVA [51] on our $\mathcal{C}ola$ dataset, we see in Table 2a that it performs below random (row o and g). This is consistent with the findings in Winoground [54].

**Training late multimodal layers from scratch help, CLIP+*MM-Adapter* performs better overall:** We improve performance by training multimodal layers from scratch on top of CLIP and FLAVA as shown in Table 2a, row e, f, j, k. Interestingly, as shown in Table 2a, linear probing or tuning the pre-trained multimodal layers of FLAVA hurts performance (row g vs row h and i). This could be because tuning adversely perturbs the parameters trained on large-scale data. Finally, CLIP+MM-Adapter (row f) performs comparably well as tuning multimodal layers on FLAVA (row f vs j and k).

## 6.3 Effect of fine-tuning data

**Difference between free-form captions and templated $\mathcal{C}ola$ queries is minimal. Hence, $\mathcal{C}ola$ templated queries are useful:** In Table 2b (row a vs c, d, e), we see that templated queries perform

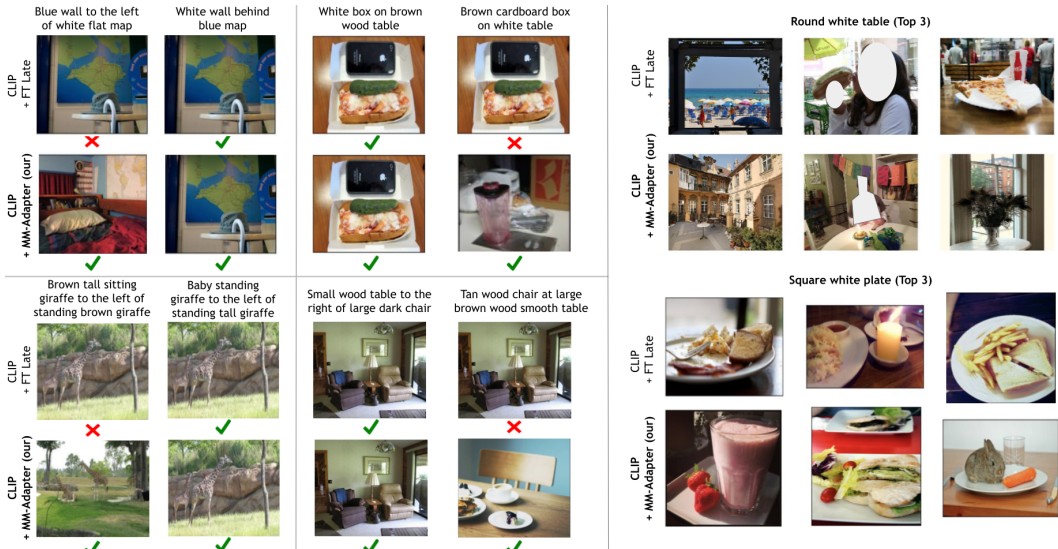

Figure 3: Qualitative results of multi-object matching (left) and retrieving a single object with multiple attributes (right).

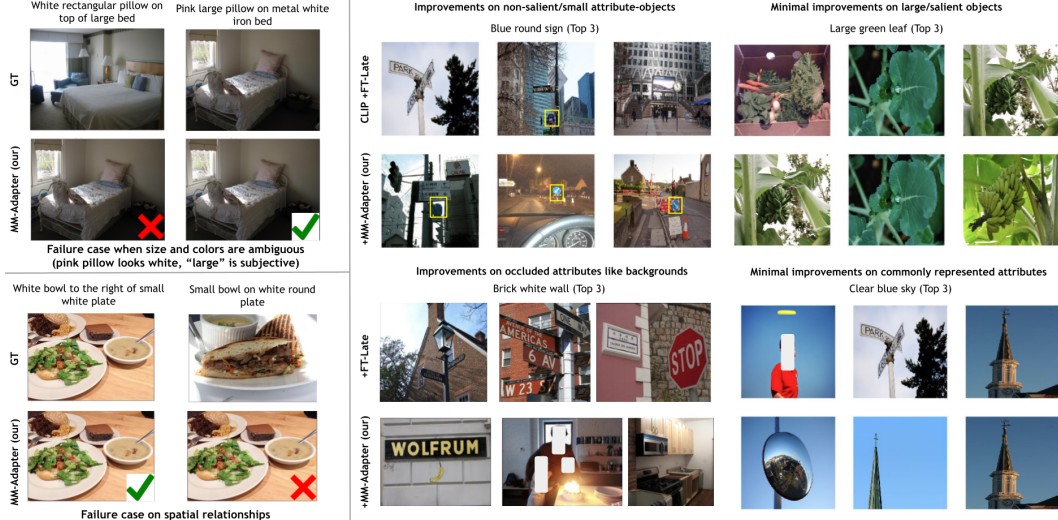

Figure 4: Qualitative results on cases where models struggle with multiple object-attribute compositionality (left). Cases where we see the most improvement and the least on single-object compositional retrieval are shown on the right.

just as well as free-form region descriptions. This shows that $\mathcal{C}ola$ queries are still useful for attribute-object binding despite not being free-form.

**Having multiple objects with multiple attributes in a caption helps:** When we combine single object captions with multiple object captions, we see in Table 2b that performance increases especially on the multi-object $\mathcal{C}ola$ setting while maintaining performance on the single-object setting (row b vs c). The multi-object and single-object captions are formed on the same number of images. Combining all types of data along with hard negatives (row e) doesn't seem to affect performance much.

### 6.4 Quality of $\mathcal{C}ola$ Benchmark

**Our $\mathcal{C}ola$ mAPs are harder and are more difficult to improve on:** We also computed the performance based on the standard formulation of mAP, where all images are included in the list to

retrieve from. In contrast, $\mathcal{C}ola$ mAP only includes hard distractors as defined in Sec. 3. We see that trends remain the same with the standard mAP. However, our $\mathcal{C}ola$ mAP is harder to improve on. The standard MAP (supplementary, Table 2) improves by more than 2x on GQA and 10x on CLEVR. In contrast, we only improve by 1.09x on GQA and 2x on CLEVR with our harder $\mathcal{C}ola$ MAP.

**Ambiguous colors, spatial relationships, and size are some common themes where models underperform in $\mathcal{C}ola$ benchmark:** We analyze the types of compositional queries that models find difficult on our benchmark. In the qualitative examples shown in Fig. 4, we see that compositions involving ambiguous colors (due to lighting) and spatial relationships are difficult for multiple-object cases. This is likely because spatial relationships are often inconsistently annotated in training data - *e.g.* "to the left of" can sometimes be from the viewer or image perspective. More examples are in the supplemental (Fig. 4, 16, 17).

**Significant improvements on non-salient/occluded objects. Improvements on larger objects are minimal:** On attribute-object compositional retrieval for single objects, we see the most improvements using our adaptation method are on non-salient or occluded objects (like a small sign), as shown in Fig. 4 (right). Queries that are commonly represented in training sets (like "clear blue sky" - skies are most commonly blue) have minimal improvements from pre-trained models.

## 7    Discussion and Limitations

This work finetunes vision-language models to test design choices for compositional attribute-object retrieval. Thus, compared to the original pre-trained models [42], we may lose some other generic capabilities, such as question answering, captioning, etc. For example, our best adaptation scores 83% zero-shot on CIFAR10 [29], as compared to 87% for pre-trained CLIP. However, by just using 5% of CIFAR training data (less than 1 epoch), we reach 92% accuracy while maintaining performance on our $\mathcal{C}ola$ task. We would also like to stress that our goal is not to train a new foundation model but to explore design choices that allow for compositional reasoning. While we focus on attribute-object compositionality, there is still significant room for exploration of other types of compositional structures such as relationships, scene graphs, and counting [14, 12]. A great avenue for future work could be collecting more detailed annotations (about the type of objects and compositions in the queries) on larger sets of images to help pinpoint themes where models are failing. Further testing is also required to see how it fares with sensitive attributes and objects— whether it is predisposed towards attaching incorrect attributes to objects because of racial/political biases in the data [6]. Additionally, it is important to re-evaluate our results in the context of newer vision-language models that are being proposed. Finally, we would like to point out that our finetuning strategy isn't specific to compositions, suggesting its potential applicability for adaptation to other downstream tasks, especially since similar strategies and modules have been used for other computer vision tasks [52, 51, 25].

## 8    Conclusion

We present a new task, $\mathcal{C}ola$, to test the compositional attribute-object binding of vision-language models. This is important for various practical applications like an assistive agent requiring an understanding of fine-grained differences between objects in cluttered workplaces. We explore the architectural choices of adapting large vision-language models that encourage such reasoning. We show that a light-weight multimodal adaptor can improve this capability in a pre-trained vision-language model as a strong baseline for further research. We hope that $\mathcal{C}ola$ serves as a strong benchmark and our adaptation choices as strong baselines for improving compositional vision-language intelligence.

**Acknowledgements.** We wish to thank Dhruv Mahajan and Jang Hyun (Vincent) Cho for their valuable guidance in the initial phases of the project. We also wish to thank the anonymous reviewers for their thoughtful comments and suggestions. This material is based upon work supported, in part, by DARPA under agreement number HR00112020054 awarded to Kate Saenko and Bryan Plummer at BU. Any opinions, findings, and conclusions or recommendations expressed in this material are those of the author(s) and do not necessarily reflect the views of the supporting agencies.

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
