# Supplemental: A Benchmark for Compositional Text-to-image Retrieval

[1]**Arijit Ray**      [2]Filip Radenovic      [2]Abhimanyu Dubey      [1]Bryan A. Plummer
[2,3]Ranjay Krishna      [1,2]Kate Saenko

{array, bplum, saenko}@bu.edu, {filipradenovic, dubeya}@fb.com,
ranjay@cs.washington.edu

[1]Boston University, [2]Meta AI (FAIR), [3]University of Washington

## 1   More Data Details

Our data is publicly available at https://github.com/arijitray1993/COLA. Recall that our data is built on top of four publicly available datasets. We provide some more details on how we curated our data.

### $\mathcal{Cola}$ **Single Object Data**

**GQA** GQA has annotations of objects and attributes in images. We use this to construct queries like "square white plate". We ignore bounding boxes. For test, we filter the images with at least 2 attributes per object annotation in their test split. We are left with 1952 images and 320 queries on our test set. To create a challenging set where queries are unseen, we take 150 attribute object tuples from the 320 queries from the test set that are seen the least in the training set and remove those images and queries from the training set completely. This way, we end up with 150 unseen queries with the least impact on the training set size. We report all numbers on this test set of 150 unseen and 170 seen queries. We train on the GQA train split (with the test unseen queries and corresponding images removed). Hence, we have around 67K training images and 27K queries. The number of paired examples is 450K image-text pairs.

**CLEVR** On CLEVR, we test on 96 classes on 22,500 images. We use their compositional splits. We train on condition A as described in their paper and dataset website and test on condition B. In these two splits, cubes and cylinders have unseen color and size compositions. However, for spheres, all colors and shapes are seen. Since MAP is sensitive to the number of classes, we keep the number of classes for seen and unseen the same. Hence, we leave out the spheres when reporting seen vs unseen. However, for all MAP, we report including spheres. Hence, we have 32 unseen classes, 32 seen classes and 96 classes for "all". For training, we have 168 possible queries (with colors swapped for cubes and cylinders from those in the test set) on 70K images.

**PACO** The PACO (9) dataset has 55 attributes annotated on 75 object categories on 9443 images on the test set. Since all combinations of objects and attributes would result in an intractable amount of possible compositions, we sample the 400 occurring multiple attribute-object compositions in the test set. The 400 classes are sampled by sampling the top 200 seen attribute-object queries and the top 200 unseen attribute-object queries. An attribute-object query is defined as unseen if the attributes in conjunction with that object were never a subset of the attributes in conjunction with that object in the training data. This way, we have 400 classes on 7921 images, on which we report numbers. We have 37K training images and 18K queries and 55K paired image-text examples.

### $\mathcal{Cola}$ **Multi-Obj Data**
The multi-obj data was created on the train and test splits according to the $\mathcal{Cola}$ Single-Object GQA data splits. Only the test split is cleaned using human annotations. We show some qualitative examples of the human-cleaned test set in Figure 3. We also see that some

37th Conference on Neural Information Processing Systems (NeurIPS 2023) Track on Datasets and Benchmarks.

cases remain ambiguous even after human cleaning, as shown in Figure 4. These often involve differences in perception of size (large vs small) and color (blue vs white under different lighting conditions). However, despite some minimal noise, the human accuracy of 10 independent workers on our validation set is 84%. This is opposed to our best model with an accuracy of 45%. Hence, we believe there is significant room for improvement.

**Datasheets for Datasets Answers**

We believe that the majority of the questions in the datasheets paper (2) have already been answered in the main paper. Here, we provide some additional answers. We also provide a histogram of the types of objects and attributes in our data in Figure 2

**Dataset funding agency** This project was supported by DARPA Semafor awarded to KS and BP. The findings and results reported in the paper are not the opinions of the US Government or the Department of Defense.

**Does the dataset contain all possible instances or is it a sample (not necessarily random) of instances from a larger set?** The dataset is a curated sample from larger datasets specifically aimed to test attribute-object compositionality of models.

**What data does each instance consist of?** Raw images, text captions, text-based scene graphs of objects and attributes in the image. Note that some objects and attribute annotations may be missing.

**Is any information missing from individual instances?** Yes, since it is very diffiucult to exhaustively annotate all possible objects and attributes, it is possible that some annotations are missing.

**Is the dataset self-contained, or does it link to or otherwise rely on external resources (e.g., websites, tweets, other datasets)?** Since the data is built on top of publicly available datasets, some of the annotations, like scene graphs, are linked to the external dataset.

**Does the dataset contain data that might be considered confidential** Not that we are aware of since we use publicly available data from a published dataset.

**Does the dataset contain data that, if viewed directly, might be offensive, insulting, threatening, or might otherwise cause anxiety?** Not that we are aware of.

**Does the dataset identify any subpopulations (e.g., by age, gender)?** No personally identifiable information is present in the data. We also do not conduct any analyses with sensitive attributes like race, age, sexual orientation, or religion. Some attributes like sex and hair color may be annotated in the images, but we don't explicitly analyze them since that is not the focus of the benchmark or the paper.

**Is it possible to identify individuals (i.e., one or more natural persons), either directly or indirectly (i.e., in combination with other data) from the dataset?** No personally identifiable information is present in the data. It may be purely coincidental that a person in real life may be present in the images of the data.

**Does the dataset contain data that might be considered sensitive in any way?** Not that we are aware of.

**Over what timeframe was the data collected?** Visual Genome (6) was collected in 2016. GQA (3) in 2018. PACO (9) in 2022-23. CLEVR (4) in 2016.

**Who was involved in the data collection process (e.g., students, crowdworkers, contractors) and how were they compensated?** Students ran the data collection and crowdworkers annotated the data. We do not know how much they were compensated for the datasets we build on top of. However, for our human cleaned multi-object $\mathcal{C}ola$ test set, we paid crowdworkers an average of 15 USD per hour with bonuses if they annotated examples correctly.

**Were any ethical review processes conducted?** Yes, we were exempted by IRB since the data didn't involve any personal or sensitive information.

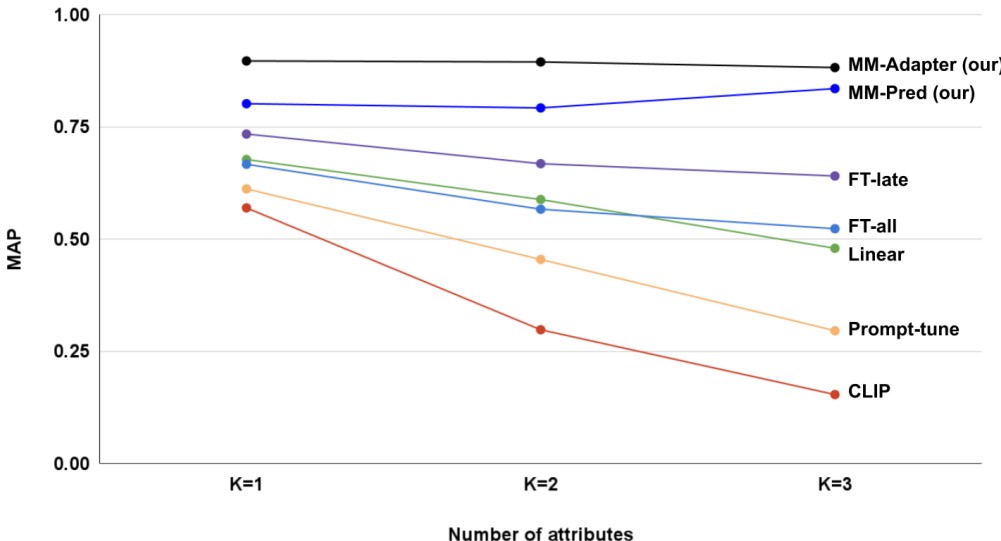

Figure 1: The MAP numbers by the number of attributes in the query on the CLEVR dataset. Note how MM-Adapter performs well even as the number of attributes is gradually increased.

**Has an analysis of the potential impact of the dataset and its use on data subjects (e.g., a data protection impact analysis) been conducted?** No, but this could be interesting future work.

**Is the software that was used to preprocess/clean/label the data available?** Yes, we shall release the code used to curate the data.

**Will the dataset be distributed under a copyright, what are the IP restrictions, and export control restrictions?** No such restrictions are present. The licenses are the same as the licenses of the datasets our benchmark is built on.

**Will the dataset be updated?** We may update the data with more examples or more annotations periodically.

## 2 More metrics and analysis

Recall, that our goal is to adapt vision-language features to improve the compositional binding of attributes to objects. Specifically, we aim to improve the classification of a query involving single or multiple objects with multiple attributes in an image. Hence, we perform some analysis to see how our performance is affected by increasing the number of attributes. Recall, that we also report numbers on our $\mathcal{C}ola$ MAP, which evaluates the model's capability to rank the images with the correct attachment of attributes to the desired object from hard distractors with incorrect attachments of attributes to objects. We also show results some other choices of MAP and the standard MAP used commonly. We show how our choice of MAP in the main paper is harder even though all trends remain the same with all choices of MAP. Finally, we also show performance from other choices of doing multimodal fusion for our MM-Adapter and MM-pred approaches, showing that this adaptation strategy holds for various other choices as well.

**Performance by number of attributes** We vary the number of attributes in the query for a single object setting and check performance with increasing attributes. The results are shown in Figure 1. We see that the baseline CLIP and finetuning has higher performance on single-attribute queries than multi-attribute queries. We show that our MM-Adapter maintains improved performance on both the single-attribute and multi-attribute cases.

**Other evaluation metrics**

**QueryAll MAP** Recall that in the main paper, we compute MAP among hard distractors for the $\mathcal{C}ola$ single-object setting. The hard distractors are images that have *any* of the attributes and object

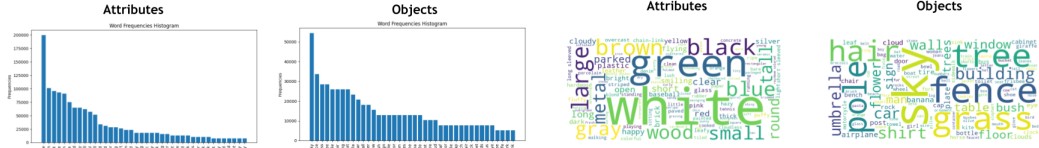

Figure 2: Overview of the types of attributes and objects in our data. They correspond to practical objects in daily life.

|  | QueryAll MAP ↑ | | | Mean Rank | |
|---|---|---|---|---|---|
|  | All | Seen | Unseen | ✓ ↓ | × ↑ |
| CLIP+linear | 49.56 | 48.21 | 31.78 | 21.78 | 54.17 |
| prompt-tune | 31.07 | 29.34 | 29.43 | 31.37 | 52.72 |
| FT-all | 54.58 | 52.05 | 21.27 | 21.06 | 54.32 |
| FT-late | 66.16 | 70.15 | 30.99 | 14.03 | 55.35 |
| MM-Pred (our) | 85.51 | 77.85 | 81.18 | 9.69 | 55.99 |
| **MM-Adapter (our)** | **90.35** | **81.95** | **90.53** | **8.85** | **56.12** |

Table 1: Two other choices for a hard metric computed on the CLEVR (4) dataset.

words in the query. The model needs to rank the images with the correct attachment of the attributes to the *desired* object (as opposed to simply existing somewhere) to achieve a higher MAP. Here, we design another similar hard MAP. Here, we restrict the list of images in the pool to have *all* the query attributes and objects. Hence, for a query "cyan metal cylinder", we rank among images that have "cylinders" AND "cyan" objects AND "metal" objects. In the main paper, the MAP uses an OR instead of an AND operation. The results are shown in Table 1 and we observe that all trends remain the same with this metric as presented in the main paper. However, we observe that that this metric can only be applied to CLEVR since annotations are exhaustive. In real datasets like GQA, the number of such annotated hard distractors is limited; hence, we do an OR operation to keep a high number of images to rank from. When applied to a dataset like GQA, the trends are the same, but the numbers are spuriously high since there are very few distractor images.

**Mean Rank of GT (✓) vs distractors (×)** Based on the hard distractors we made for the QueryAll MAP above, we also report the mean rank of the images with the correct attachment of attributes to the object versus the mean rank of the images with the wrong attachment of attributes. The results are also shown in Table 1. We observe that all trends remain the same, so report only one of them in the main paper.

**Standard MAP** In contrast to our hard $\mathcal{C}ola$ MAP's, we also compute the MAP on all images in the validation set regardless of hard or easy distractors. Once again, we see all trends remain the same as shown in Table 2. However, we note that the MAP numbers on all images are much lower. This is becuase of two reasons - a) the number of images to rank from is higher, and b) datasets like GQA have missing annotations, hence there are many images that get denoted as a negative retrieval becuase of a missing annotation. When we restrict the images to at least have one of the query words, this noise reduces somewhat. However, note how it is easier to improve on this overall MAP than on the harder $\mathcal{C}ola$ MAP reported in the main paper. This shows models can quickly improve on distinguishing coarse-grained differences but differentiating between the fine-grained ones (as evaluated by the $\mathcal{C}ola$ MAP) is harder.

**F1 Score** We also ran a sample of the evaluation using F1 on the GQA split of COLA single-objects, and we see that all trends remain the same when comparing F1. For instance, F1 of CLIP baseline is 0.28, whereas FT-all is 0.31, FT-Late is 0.33, and our MM-Adapt is 0.39, MM-Pred is 0.40. Our conclusion stays the same: adapting the multimodal attention layers is better than tuning the split-modal attention layers (FT-Late), fine-tuning the entire model, or linear probing.

|  |  | Overall MAP | | |
|---|---|---|---|---|
|  |  | **All** | **Unseen** | **Seen** |
| **GQA (3)** | CLIP | 0.65 | 0.35 | 0.91 |
|  | + prompt-tune | 8.72 | 9.63 | 7.91 |
|  | + Linear probe | 12.81 | 13.29 | 12.52 |
|  | + FT all | 11.47 | 10.87 | 11.96 |
|  | + FT late | 13.39 | 13.70 | 13.10 |
|  | **+ MM-Pred (our)** | 16.76 | 17.45 | 16.15 |
|  | **+ MM-Adapter (our)** | 17.40 | 16.79 | 17.95 |
|  | FLAVA | 7.33 | 6.43 | 8.15 |
|  | + FT-late | 9.71 | 9.49 | 9.90 |
|  | **+ MM-Pred (our)** | 17.68 | 19.24 | 16.29 |
|  | **+ MM-Adapter (our)** | 20.03 | 20.70 | 19.42 |
| **CLEVR (4)** | CLIP | 6.42 | 6.36 | 6.29 |
|  | + prompt-tune | 29.42 | 23.02 | 27.79 |
|  | + Linear probe | 47.83 | 29.33 | 46.54 |
|  | + FT all | 51.99 | 18.40 | 47.63 |
|  | + FT late | 63.93 | 27.20 | 67.00 |
|  | **+ MM-Pred (our)** | 83.40 | 76.82 | 76.10 |
|  | **+ MM-Adapter (our)** | 88.15 | 89.40 | 76.90 |
|  | FLAVA + linear | 18.76 | 16.77 | 17.82 |
|  | + FT-late | 77.59 | 71.91 | 66.25 |
|  | **+ MM-Pred (our)** | 90.41 | 85.74 | 86.05 |
|  | **+ MM-Adapter (our)** | 91.08 | 86.60 | 87.39 |
| **PACO (9)** | CLIP | 0.71 | 0.11 | 1.31 |
|  | + prompt-tune | 6.19 | 2.78 | 9.61 |
|  | + Linear probe | 8.22 | 3.83 | 12.61 |
|  | + FT all | 7.19 | 3.00 | 11.38 |
|  | + FT late | 9.29 | 5.37 | 13.21 |
|  | **+ MM-Pred (our)** | 9.63 | 4.00 | 15.26 |
|  | **+ MM-Adapter (our)** | 10.00 | 6.50 | 15.22 |
|  | FLAVA + linear | 3.45 | 1.73 | 5.17 |
|  | + FT-late | 6.31 | 2.00 | 10.60 |
|  | **+ MM-Pred (our)** | 10.77 | 4.77 | 16.76 |
|  | **+ MM-Adapter (our)** | 12.02 | 6.36 | 17.67 |

Table 2: Standard MAP on all images with multiple attributes on objects annotated in the test set (not just hard distractors like our $\mathcal{C}ola$ MAP). Note how we can improve significantly (eg, 0.65 to 17.40 on the GQA split - 10x), but by a much lesser fraction on our $\mathcal{C}ola$ MAP which is only among hard distractors.

|  |  | | $\mathcal{C}ola$ Single-Obj MAP | |
|---|---|---|---|---|---|
|  |  | Params | All | Unseen | Seen |
| **GQA (3)** | Unimodal | 13M | 42.19 | 44.61 | 40.01 |
|  | FLAVA | 9.9M | 47.43 | 48.95 | 46.05 |
|  | ALBEF | 10.9M | 45.2 | 48.23 | 42.6 |
|  | MDETR | 7.8M | 46.83 | 48.86 | 44.99 |
|  | FIBER | 15M | 47.08 | 50.01 | 44.43 |
|  | FIBER-MM | 12M | 46.05 | 48.91 | 43.47 |
| **CLEVR (4)** | Unimodal | 13M | 64.05 | 27.53 | 67.48 |
|  | FLAVA | 9.9M | 88.21 | 89.52 | 77 |
|  | ALBEF | 10.9M | 85.56 | 85.47 | 72.97 |
|  | MDETR | 7.8M | 89.35 | 89.4 | 80.2 |
|  | FIBER | 15M | 82.9 | 76.97 | 73.5 |
|  | FIBER-MM | 12M | 86.6 | 88.56 | 72.98 |
| **PACO (9)** | Unimodal | 13M | 15.66 | 8.74 | 22.58 |
|  | FLAVA | 9.9M | 18.56 | 11.47 | 25.66 |
|  | ALBEF | 10.9M | 18.22 | 10.57 | 25.8 |
|  | MDETR | 7.8M | 19 | 11.13 | 26.87 |
|  | FIBER | 15M | 12.34 | 5.34 | 19.35 |
|  | FIBER-MM | 12M | 11.83 | 4.49 | 19.17 |

Table 3: Different choices for multimodal fusion inspired from ways researchers have done multimodal fusion in literature. Note that these are *not* numbers from the models proposed in their papers, but the accuracy of using the style of multimodal fusion, which we use on top of frozen CLIP features. Most multimodal variants perform better than tuning similar or more number of parameters on unimodal attention layers. The main paper numbers are from the MDETR-style multimodal fusion.

## 2.1 Other choices of multimodal fusion

In our MM-Adapter and MM-Pred approaches, we use multimodal fusion. There are various ways to do multimodal fusion. Some of the salient choices are inspired by FLAVA (10), ALBEF (7), MDETR (5), and FIBER (1). We describe some of the ways we try multimodal fusion:

– **FLAVA**-inspired - self-attention on a `[CLS]` token concatenated with image patch and text tokens-Here, we take the image patch features and text token features and employ self-attention transformer (11) on the concatenated image, text and `[CLS]` tokens.
– **MDETR**-inspired - self-attention over image patch and text tokens and then, a `[CLS]` token cross attending to the image and text tokens- In the MDETR (5) paper, they use self-attention over image and text features and then multiple task tokens that cross attend to the self-attended image-text features for various tasks. Since, we have only one task here, which is retrieval, we use one `[CLS]`. We have also experimented with using multiple (100) `[CLS]` tokens to see if they learn different things. We observe that all the `[CLS]` tokens learn the same thing with minimal performance gap. This is the choice of multimodal fusion that we report in the paper for both our MM-Adapter and MM-Pred approaches.
– **ALBEF**-inspired - text cross-attends to image - Here, first we have separate unimodal self-attention layers on the image patch and text token features. Then, the text token features cross-attend to the image patch features along with a `[CLS]` token. The `[CLS]` output is then used for MM-Pred (prediction using fully-connected layer) or MM-Adapter (cosine similarity to frozen text features).
– **FIBER**-inspired - text cross-attends to image and vice versa- Here, first we have separate unimodal self-attention layers on the image patch and text token features. Then, have text token features cross-attend to the image patch features along with a `[CLS]` token. We also have the image patch features cross-attend to text token features along with another `[CLS]` token. We finally measure the cosine similarity of the two `[CLS]` tokens.
– **FIBER-MM** - In the above FIBER and ALBEF style fusion, we used separate unimodal self-attention layers on the image patch and text token features before the cross attention. Here, we design a modification, we use a multimodal self-attention on the image patch and text tokens first, like FLAVA. Then, we do cross-attention like FIBER as described above.

Accuracies on GQA, CLEVR and PACO for $\mathcal{C}ola$ single-object case on the above-described multimodal choices are shown in Table 3. We see similar trends as the choice of multimodal attention reported in the paper. All the methods of doing multimodal fusion work better than unimodal fusion. Also, while some choices work better than others, note how using the multimodal layers as a feature adaptor (MM-Adapter) works better than using it as a prediction head (MM-Pred) for all design choices.

## 3 Implementation details

Now, we present more implementation details of the models and adaptation strategies used in the main paper. We also provide more details on the datasets used.

### 3.1 Model architecture details

Recall that we have use a CLIP (8) image and text encoder to extract image and text region features. Here are some additional details for each of the choices of adaption we tried:

– **Linear**: We train a linear probe on the pre-trained representations. We train a separate linear layer on top of the image and text pooled features for CLIP (8). Each linear layer transforms the 512-dimensional image and text representation to another 512-dimensional embeddings. Finally, we compute the cosine similarity between the two transformed embeddings.
– **Prompt-tune**: We tune the embedding layer of the text words used in our training queries while keeping everything else frozen.
– **FT all**: We fine-tune the whole model. This involves tuning 151M parameters in the case of CLIP.
– **FT Late**: We take the second-last layer features from the image and text encoders of CLIP. There are 49 image patch features and K text token features (K depends on the input query length, but it is capped to 77). We train a separate transformer encoder layer on the 49 image patch embeddings and the K text tokens. The transformer encoder has 2 transformer encoder self-attention layers with 4 heads each. We tried variations of 1 layer, 2 layers and 3 layers and report the best performance. This design is chosen to be the most similar in the number of parameters and approach to our multimodal adaptation approach to be a strong baseline.
– **MM-Pred**: Here, we use multimodal attention as a prediction head like common multimodal models (10; 5), but train it on the frozen CLIP (8) base image and text encoders. Once again, the multimodal transformer encoder has 2 layers with 4 heads each. We predict a score using a fully-connected layer on the `[CLS]` token output of the multimodal attention that maps the 512-dimension embedding to a 1-dimensional score.
– **MM-Adapter**: This differs from our **MM-Adapter** approach, where we use multimodal attention to adapt the image representation and use their cosine similarity to the text features.

For the image-text-matching loss, we get a score for each image-text pair in a batch. For each score, we compute the binary sigmoidal cross entropy and take the average in the batch. We use a sigmoidal cross entropy since for each image, there can be multiple text queries that are true and vice versa. We train using a learning rate of 0.00001 and a weight decay of 0.0001 for the models on top of CLIP. For adaptations on top of FLAVA, we see that we need a higher learning rate to converge quicker, hence, we use a learning rate of 0.001 and a weight decay of 0.0001.

## 4 Qualitative results

**Single-object case** Figures 5, 6, 7, 8 show examples of top 5 retrievals based on common adaptation methods and our MM-Adapter method on the $\mathcal{C}ola$ single-object setting on the GQA (3) dataset. Each row in the image is a different adaptation method (based on the methods shown in Table 1 in the main paper). Note how we improve on multiple attributes attached to non-salient and small objects. Figures 9, 10, 11 show some cases where we see marginal improvements from off-the-shelf CLIP or simpler adaptation techniques like fine-tuning or linear probing. We observe that marginal improvements are mostly on queries with large areas of the image like sky and water. The existing CLIP (8) model is fairly good at such large salient objects, especially when paired with common attributes like "green" for the object "leaf".

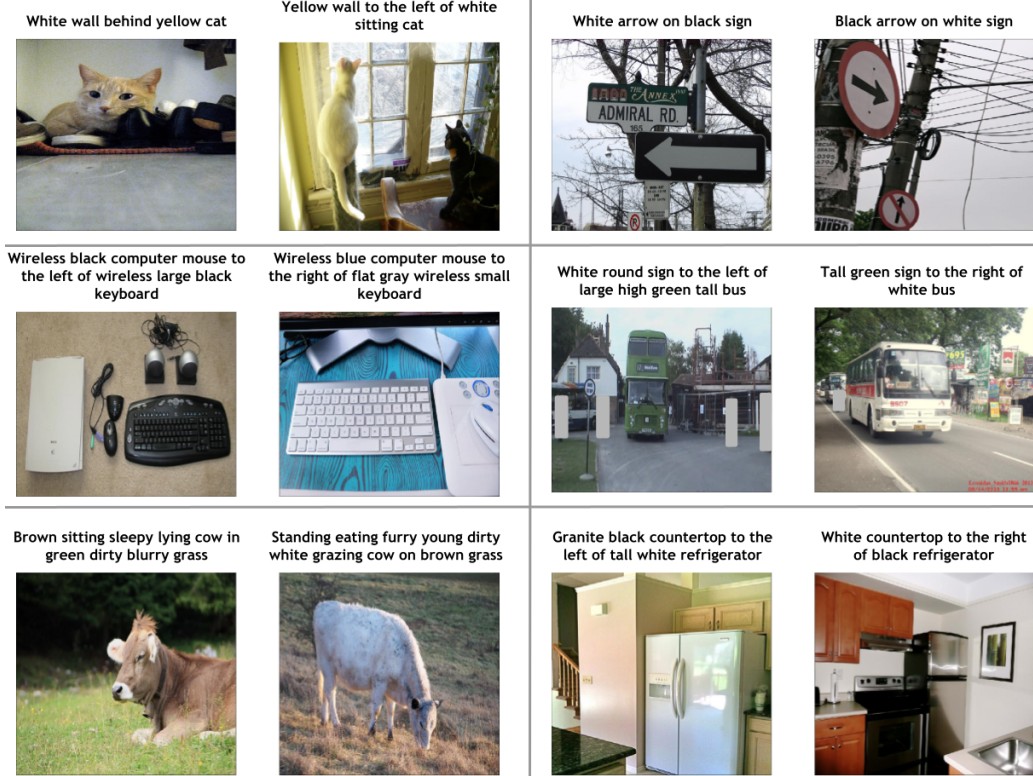

Figure 3: Some examples from the $\mathcal{C}ola$ multi-obj setting.

**Multi-object case** Figures 12, 13, 14, and 15 show some results on the $\mathcal{C}ola$ multi-object setting. Similar to the observations in the single-object setting, we improve the attribute-object binding capability even when the objects are non-salient in the image. In addition to relational compositionality, as shown in Figures 16 and 17, our method also fails to understand fine differences in the relative strength of attributes and when objects are occluded to a high degree.

All images we use are from publicly available datasets, and we are unaware of any correspondences with identifiable humans in real life.

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

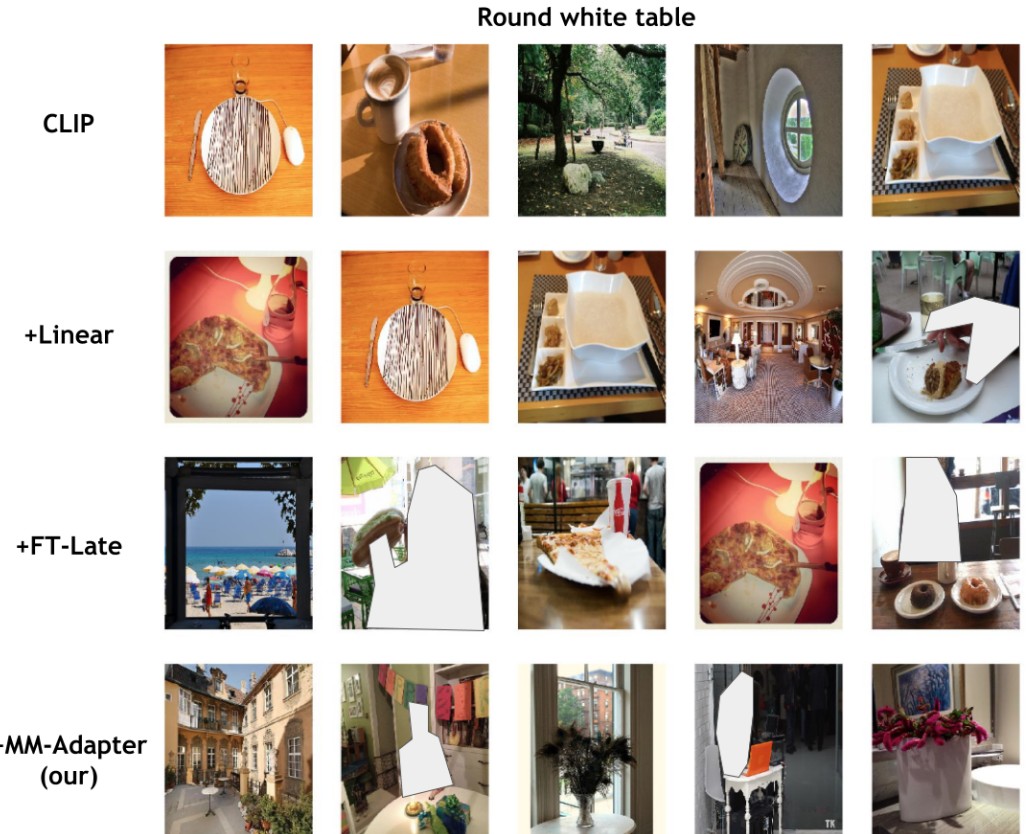

Figure 5: Qualitative results on multiple attributes attached to an object. Note how we improve on many attributes attached to small non-salient objects in the cluttered scene. The round white table in the test images were often small and hence, the original model had trouble finding them. Note how the original CLIP only find the slaient black metal chair (first row), and in comparison, we find smaller non-salient ones as well (last row)

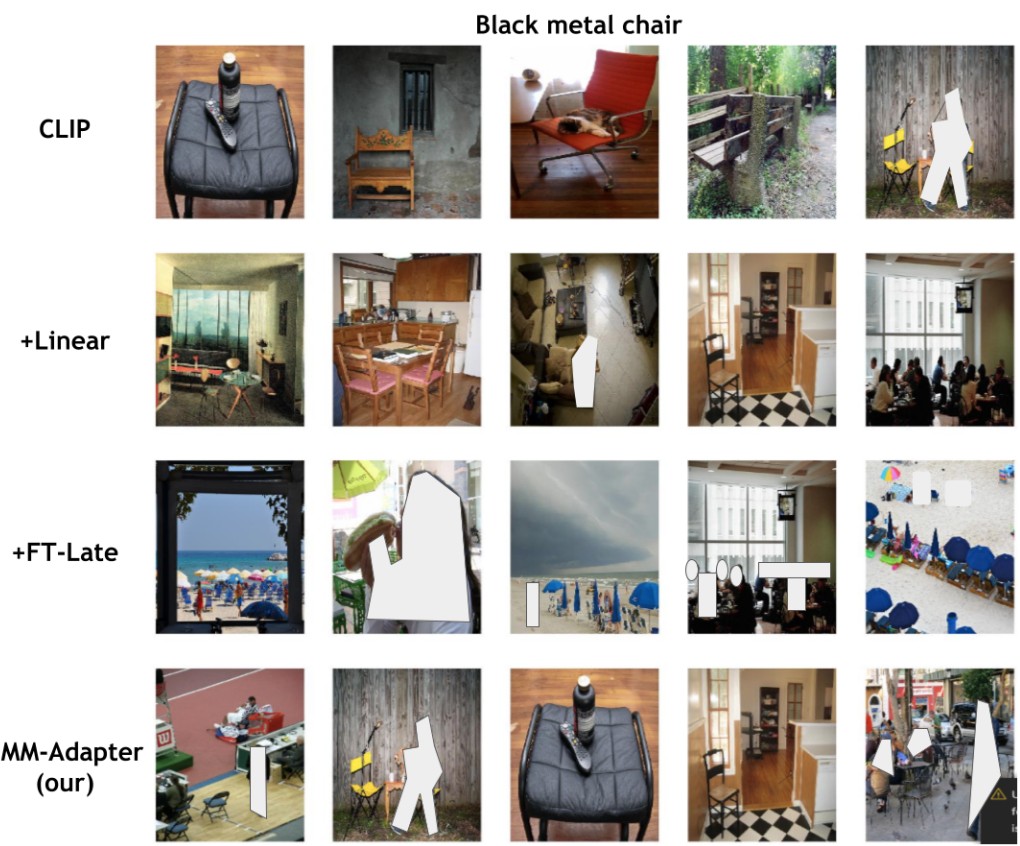

Figure 6: Qualitative results on multiple attributes attached to an object. Note how we improve on many attributes attached to small non-salient objects in the cluttered scene.

**Blue round sign**

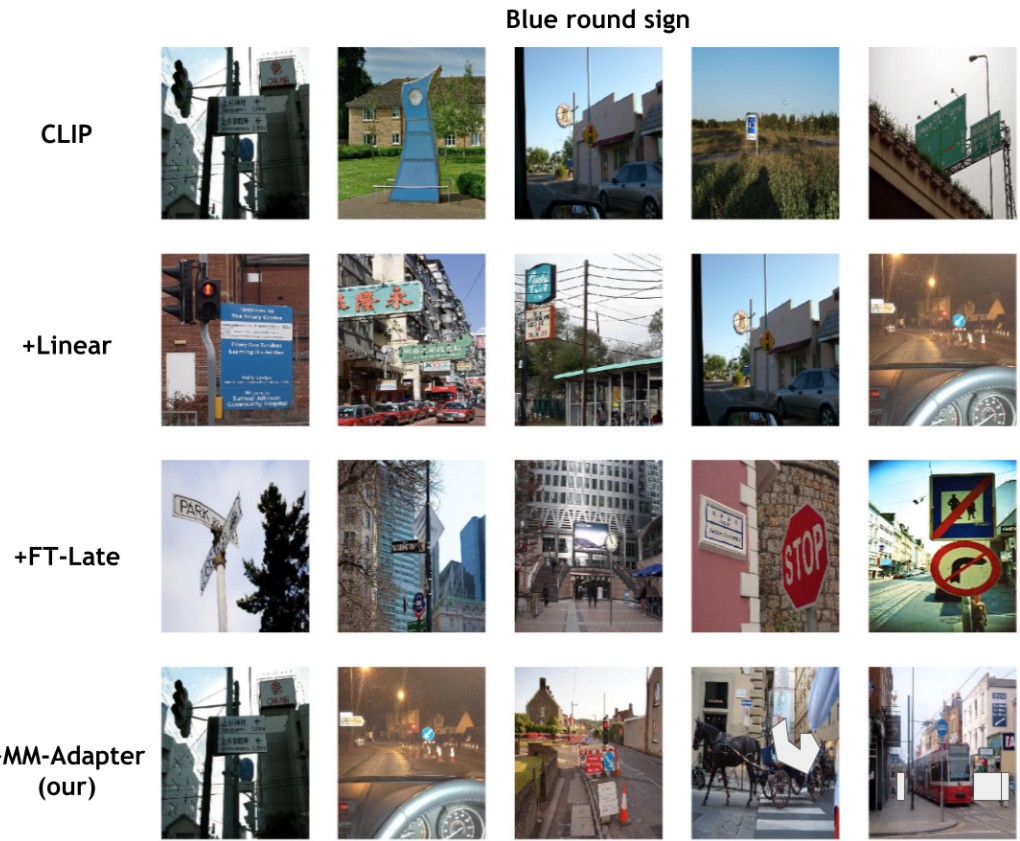

CLIP

+Linear

+FT-Late

+MM-Adapter
(our)

Figure 7: Qualitative results on multiple attributes attached to an object. Note how we improve on many attributes attached to small non-salient objects in the cluttered scene.

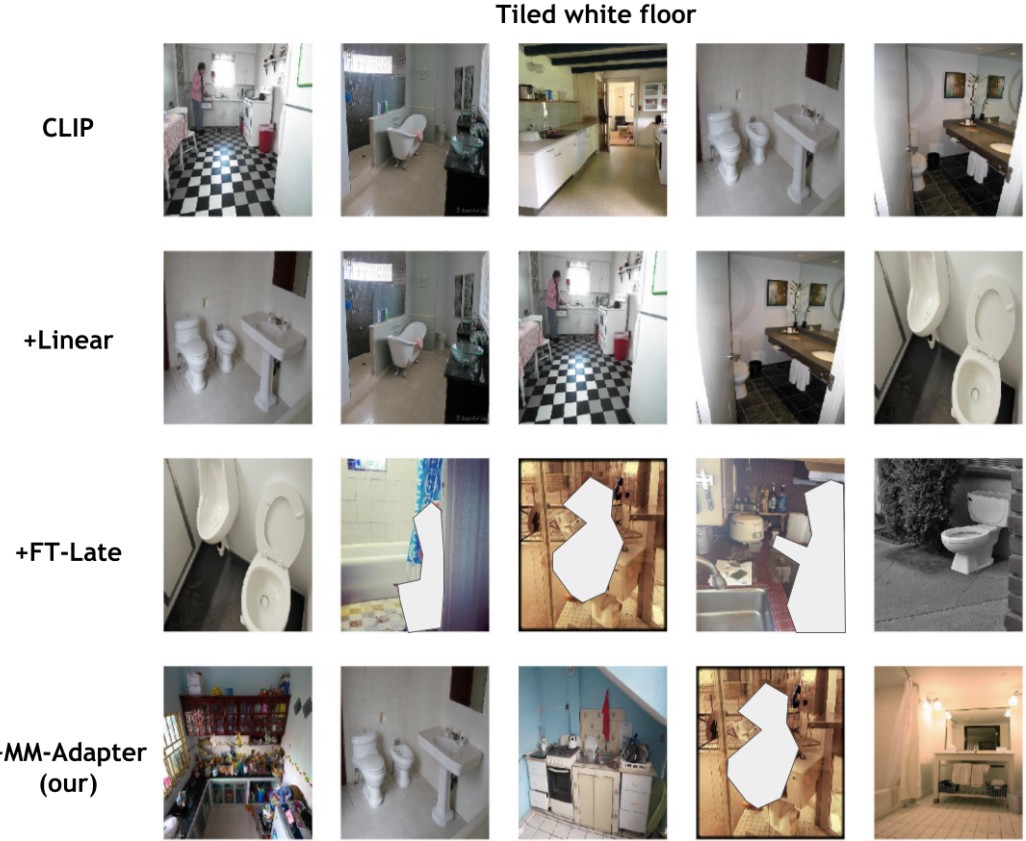

Figure 8: Qualitative results on multiple attributes attached to an object. Note how we improve on many attributes attached to small non-salient objects in the cluttered scene.

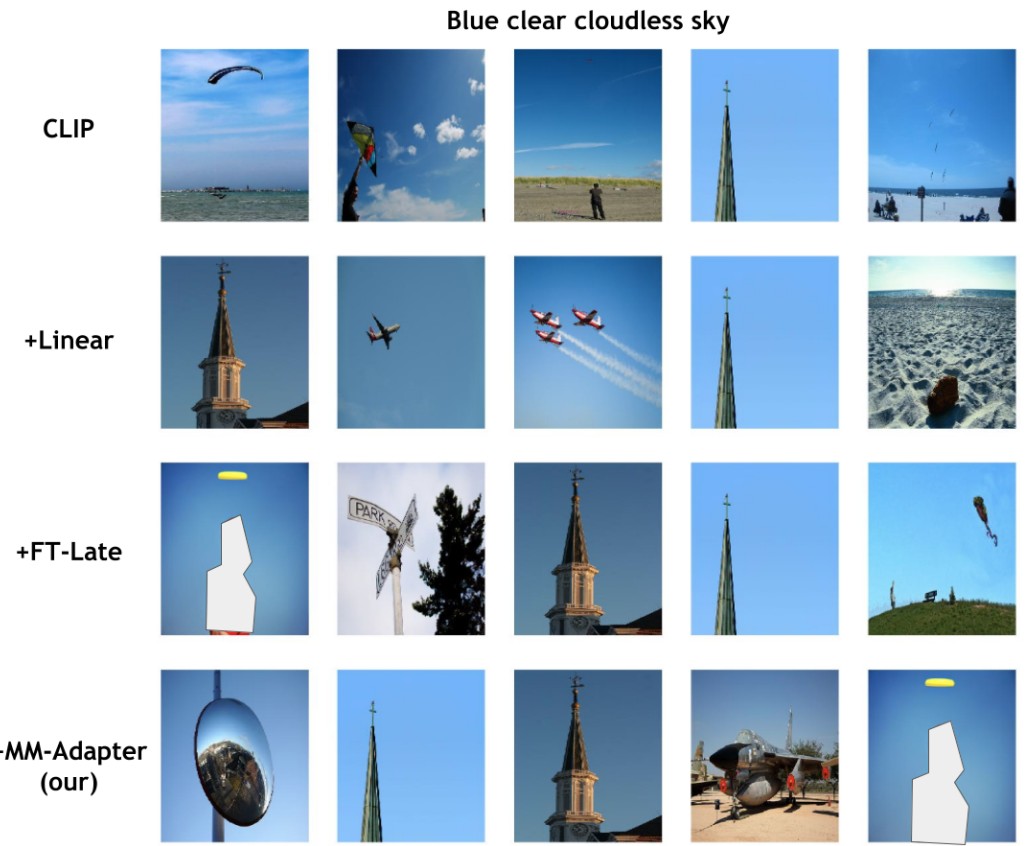

Figure 9: Queries with attributes that cover a wide area with common attributes, like blue sky, have minimal improvements from off-the-shelf or simple adaptation strategies since existing models perform well on such queries already.

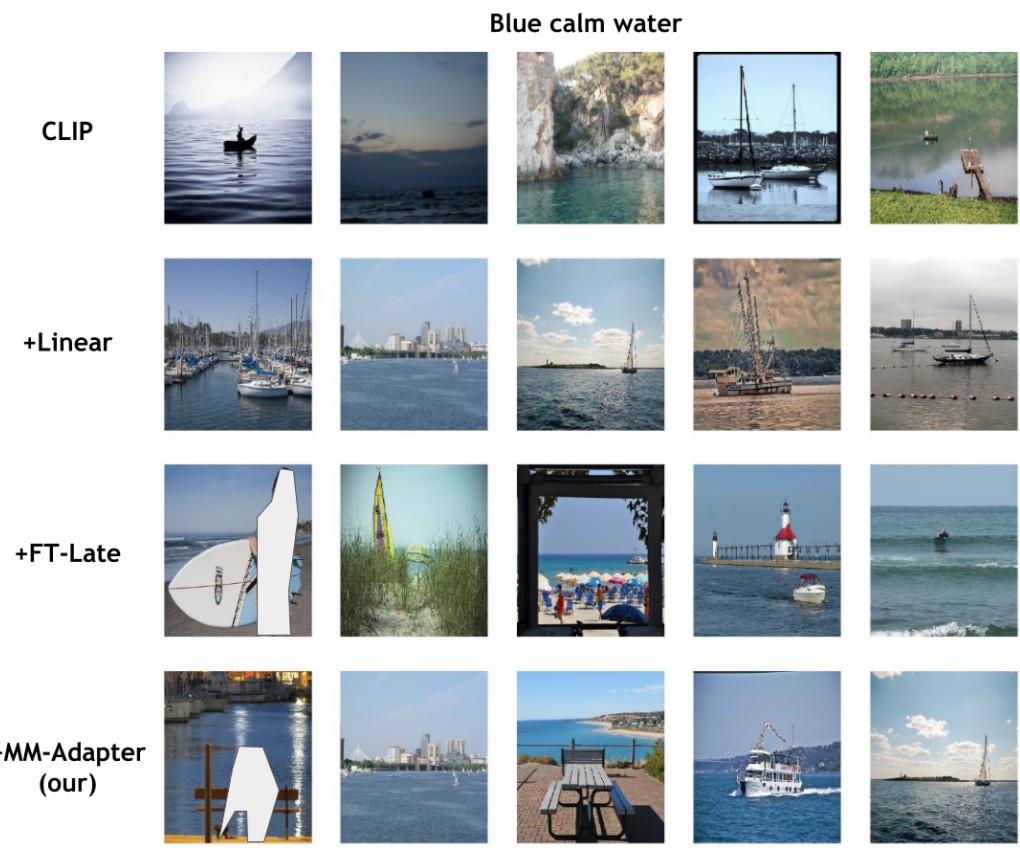

Figure 10: Queries with attributes that cover a wide area, like water bodies, have minimal improvements from off-the-shelf or simple adaptation strategies since existing models perform well on such queries already.

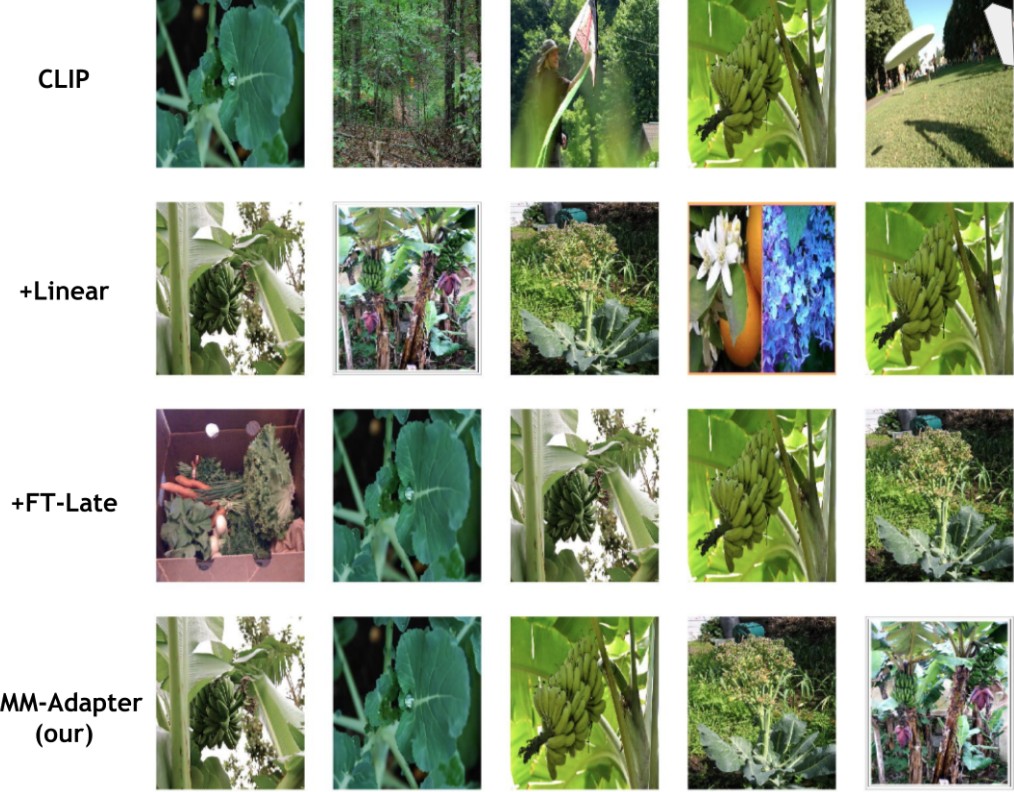

Figure 11: Queries with attributes that cover a wide area with common attributes, like a green large leaf, have minimal improvements from off-the-shelf or simple adaptation strategies since existing models perform well on such queries already.

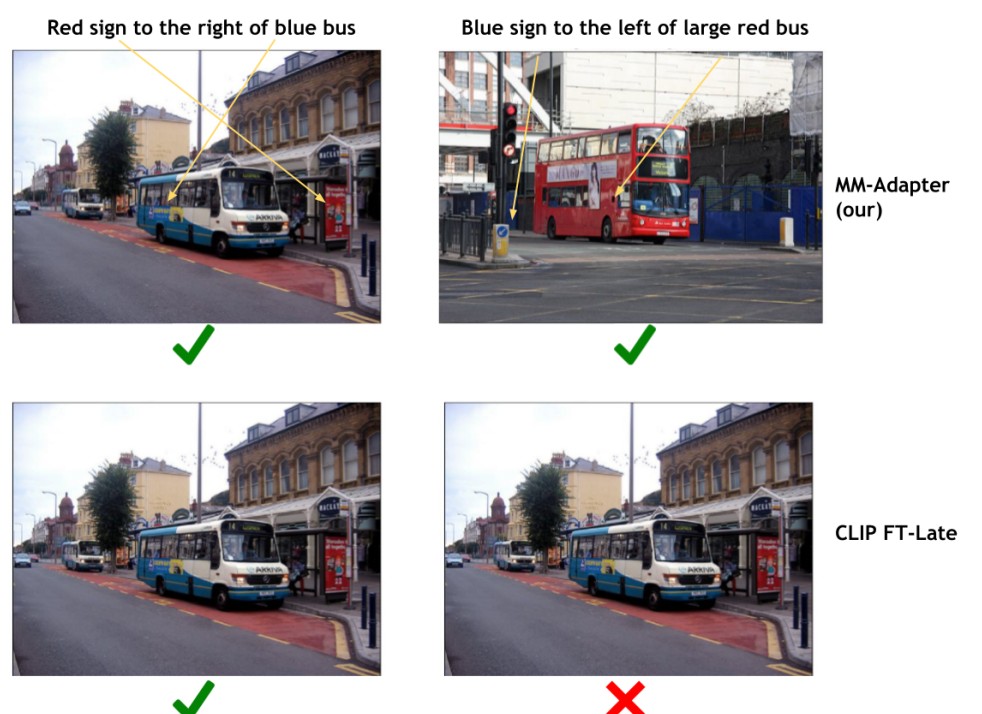

Figure 12: Qualitative results on multi-object cases. Once again, we see significant improvements on compositions involving small non-salient objects such as a small sign.

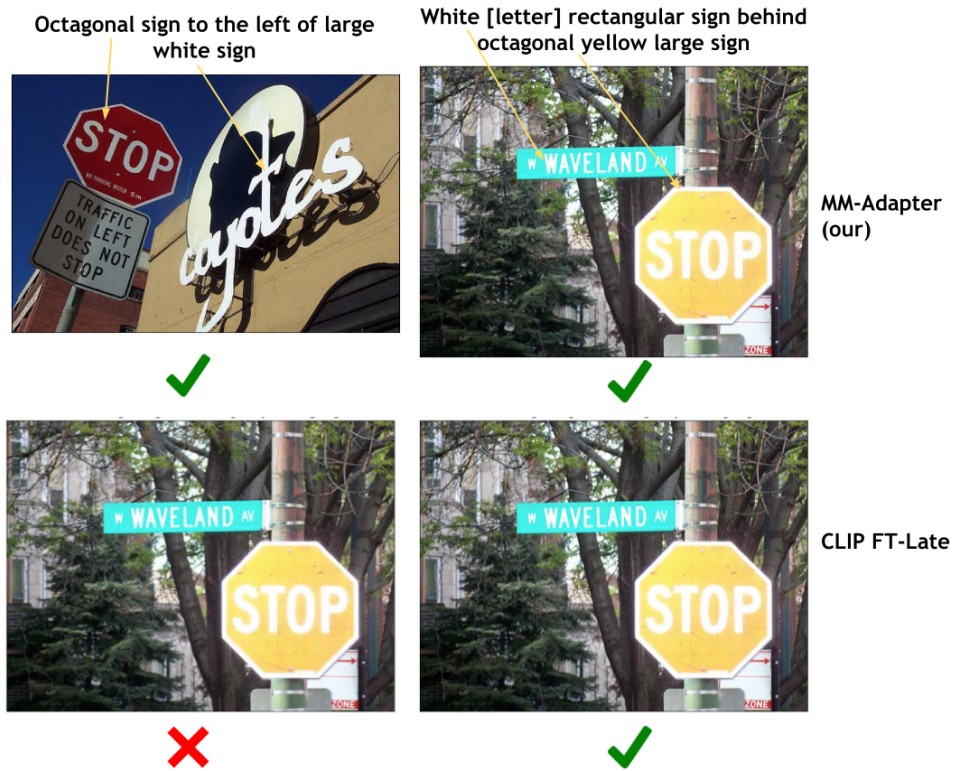

Figure 13: Qualitative results on multi-object cases.

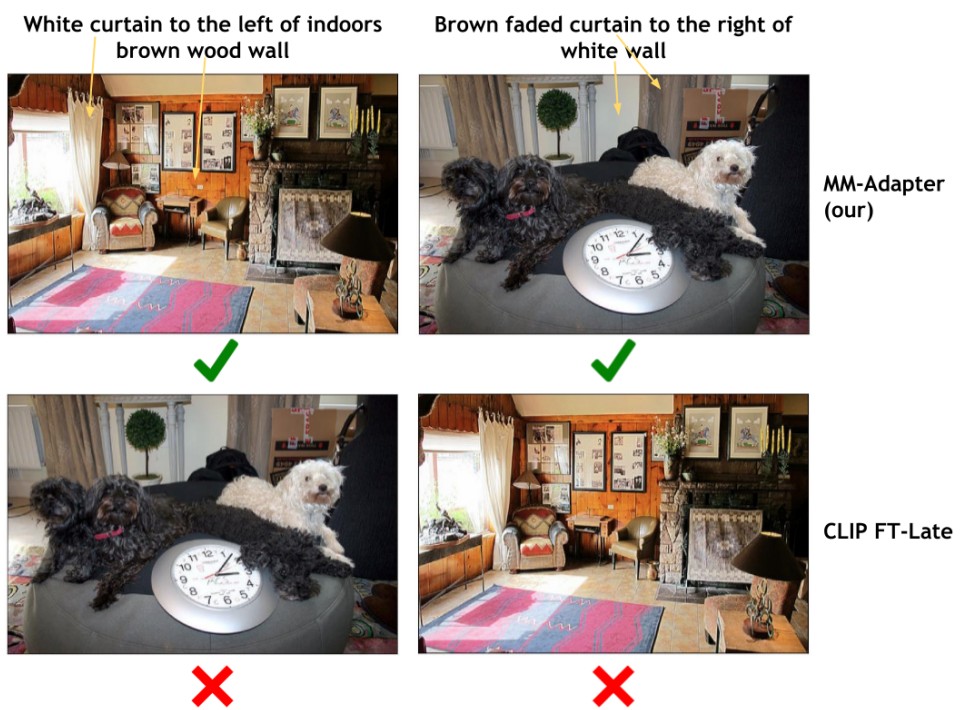

Figure 14: Qualitative results on multi-object cases. We see significant improvements on compositions where the images have a lot of clutter and distractor objects - many things are white and brown in the scenes.

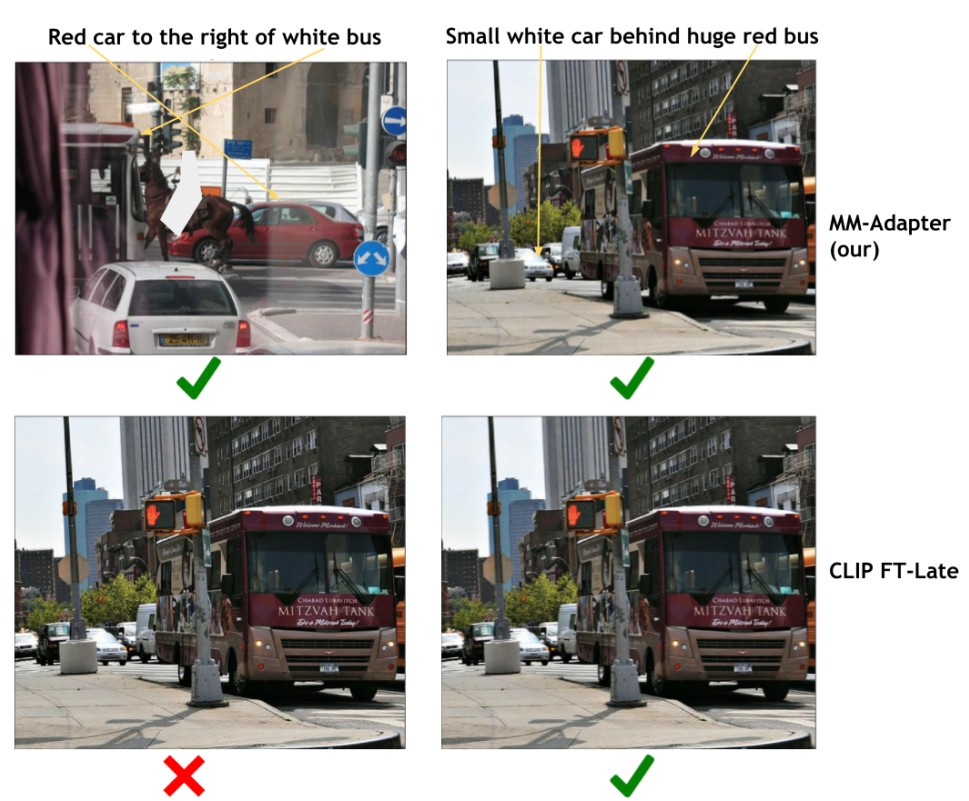

Figure 15: Qualitative results on multi-object cases.

Figure 16: Our method performs somewhat poorly on very fine-grained relative differences. In the example above, a brown chair is underneath a brown desk, but the desk is not empty. In fact, even the desk in the correct image for that caption is not technically empty, but it is more empty than the distractor and our model fails to understand the relative difference.

Figure 17: Our method also performs poorly on occluded objects or when objects have some of the attributes of the distractor as well. In the example above, the doors are not clearly in view. In addition, the brown door also has a white stripe, which further confuses the model.