# OpenReview forum: "Cola: A Benchmark for Compositional Text-to-image Retrieval"
_NeurIPS.cc/2023/Track/Datasets_and_Benchmarks — NeurIPS 2023 Datasets and Benchmarks Poster_

### Official Review · Reviewer_wAA7 · 2023-06-30
**Review of paper**

**Rating:** 7
**Confidence:** 2
**Clarity:** The paper is well-written and reasona…

**Strengths:**

The problem the authors have presented their contributions for is important for deep learning. The simplicity of the tasks, paired with the low results that the baseline produce on them, makes for a good case (which is well-accepted) that compositionality is a hard task for even recent deep learning models.

The fine-tuning strategies, being task-agnostic, presented by the authors are a good addition to the dataset the authors present.

The experimentation, while not particularly extensive, is still enough, in my opinion, to show the usefulness of the dataset the authors present.

**Additional Feedback:**

No further comments.

**Correctness:**

I believe that the experimentation is fairly correct. However, I am not fully sure of my judgement.

**Documentation:**

The authors have presented a link to their data and have stated that they would release the code they used as well.

**Ethics:**

As far as I am aware, there should be no ethical concerns from this paper that are not seen in other works on tasks such as image captioning and visual question answering.

**Limitations:**

1. The statistics regarding the dataset are largely absent in the paper. The authors could include the distribution of object-classes, the distribution of properties for the objects etc. The purpose of this is to showcase the lack of any biases that could have caused the low results. For example, choosing objects and their properties which are particularly hard for a model. Such examples would essentially show that a model is bad at working with certain object-attribute pairs, rather than actually having a lack of compositional behaviour.
2. The authors are requested to also include performance of a text-only model to showcase the absence of any further bias in the dataset.

**Opportunities For Improvement:**

The areas for improvement are as follows:

1. In order for the multi-object dataset to be relevant for the near future, it needs to be of a larger size (210 images is not enough), as models can be trained to rapidly improve on a small set. The idea behind the dataset is still valid, but needs more data.

2. While the authors experimentation (seen in terms of the number of baselines) suffices for the purposes of showcasing their claims of Cola being relevant, the readers would benefit from an increased number of baselines.

**Relation To Prior Work:**

Yes, the paper discusses previous work.

**Summary And Contributions:**

The authors introduce a dataset of image-query pairs as test for compositional understanding in multi-modal models. The authors show that this dataset, while conceptually simple, is quite challenging for several models. The authors also present new strategies for fine-tuning which allows for better performance on their dataset,

---

> ### Author Response · Authors · 2023-08-16
>
> Thank you for your positive and helpful comments. We appreciate that the reviewer finds that the contributions “makes for a good case (which is well-accepted) that compositionality is a hard task” and that our experiments suffice in showing “the usefulness of the dataset the authors present”.
>
> **The reviewer suggested a future direction with more data for the multi-object split:**
> Yes we agree that this would be a good future direction. We were limited by the availability of annotated hard distractors in the validation set of GQA. Specifically, we needed to find pairs of objects and attributes where the attributes or the objects are swapped. Finally, not all annotations were accurate and hence, after rigorous cleaning, we had 210 high quality quadruplets of image-text pairs with attributes and objects swapped.
>
> **The reviewer suggested more baselines:**
> Yes, we agree that testing more recent models on this benchmark will be useful. Our goal was to explore the architectural design choices for compositionality and hence, our choices of FLAVA, MDETR, ALBEF and FIBER covered the salient works. To show the value of adapting a cross-attention over just using a pre-trained one, we compare to a model with cross-attention pre-trained - FLAVA. Since our submission, more models have been released and should be included in future work but are out of scope for this submission.
>
> **The reviewer asked us to include a distribution of object-classes and a distribution of attributes:**
> This is a good suggestion. We computed a wordcloud and histogram of objects and attributes and will present them in our appendix.
>
> **The reviewer asked us to visualize the performance on hard objects and attributes:**
> This is a great suggestion. We have included a few of such examples in the supplementary where we show that queries related to size, closely related colors and spatial locations are especially difficult. We looked through more training examples and saw that this is because size is subjective, close colors are harder to disambiguate due to lighting differences, spatial relations are noisely annotated due to ambiguity in perspective of viewer vs image. We will move some of this analysis from the appendix to the main paper.
>
> **The reviewer suggested adding a text-only model to showcase the absence of bias:**
> Since our task is to match the corresponding text to the correct image from a set of 2 images and 2 texts, it is not immediately clear to us how we can use a text-only model. We rely on being able to calculate a similarity score between images and text. While compositional benchmarks like CREPE might be able to run a text-only model since they are retrieving the correct text from a set of distractors, we don’t have a notion of correct text amongst distractors. This is because both the texts have a corresponding image in our case for the multi-object split and we have one text to multiple images (from which the model needs to retrieve the correct images) for the single-object split.

---

### Official Review · Reviewer_H1og · 2023-07-07
**Not really a benchmark study, needs more analysis and better clarity.**

**Rating:** 6
**Confidence:** 5

**Strengths:**

The proposed approach maintains an open-world paradigm.
Their approach shows noticeable improvements over the other fine-tuning approaches.
They are studying an interesting problem and proposing a dataset that encapsulates multiple domains of object-attribute combinations and images.

**Additional Feedback:**

Not a negative part of the review, so I did not take this into consideration on my rating, just thought I would share: https://arxiv.org/abs/2304.03659
Here we see models like BridgeTower are very strong with single attribute-object comparisons and they have a architecture that uses CLIP with cross-attention extensions.

Also not taken into consideration in my rating, but another work that could be looked at is: https://openreview.net/pdf?id=KRLUvxh8uaX (published ICLR 2023). This work does similar things with fine-tuning but also maintaining original performance on other datasets. It does so without altering the architecture as well.

**I have increased my rating based on author response. The inclusion of original scores, results on CIFAR, and computational complexity is my reason.**

**Clarity:**

* Figure 2 is hard to read and understand.
* There is no discussion on the original models.
* The proposed fine-tuning approach is not very clear and the figure does not provide much detail or clarity.
* I am confused by L190, do you mean the CLS token is projected using a set of linear layers to make a final prediction? L193 is also confusing, so are you taking the [CLS] token from the self-attention between concatenated image and text features and comparing to the frozen text encoder whose output is also passed to the self-attention mechanism? Is the only difference between MM-pred and MM-adaptor whether there is a set of linear layers after the self-attention for the [CLS] token only?
* L197 should explain what those multi-modal adaptations are from the two models mentioned. I see it is in the supplementary, but a short 1-2 sentences may be useful or even mentioning more details are in the supplementary to refer to.
* L204, I am unsure what COLA MaP is as compared to traditional MaP as discussed in the Supplementary.
* There are no details on the prompt-tuning used, how many tokens for example? The method described in the supplementary is not the general understanding of prompt-tuning. For example, CoOp, CoCoOp and more use learned contextual tokens while everything else is frozen. What is described in the supplementary is not the same.
* Table 2 should be split up and the colors are very hard to see. It is also counterintuitive that red would be highest performing.
* Flava has a cross-attention module after already, are you adding an additional one? This is very unclear how it is applied to FLAVA. Are you removing its old one and just using the modality specific encoder outputs and using a different cross-attention module? Or you finetuning the cross-attention module FLAVA already has?
* L251, what is the low resource regime?
* Why is COLA harder? Some analysis on this might be useful. It might tell us what particular aspects of this problem models are struggling with.
* With the fine-tuning approaches, there can be multiple hyper-parameters used that impact the final performance. What kinds of variations were used for the more traditional fine-tuning paradigms?

**Correctness:**

I do not see any problems with correctness except with prompt-tuning. The way it is described in the supplementary does not match the approaches mentioned in the related works (for example CoOp [60]). They do not provide additional details on the prompting approach that can justify this either. This is incorrect.

**Documentation:**

There is documentation on the datasets, but there is none on the evaluation of the models chosen or the proposed approach. This makes the benchmark not very usable or reproducible.

**Ethics:**

No concerns.

**Limitations:**

There are only two model architectures, CLIP and FLAVA. These architectures are also similar or like the proposed approached (FLAVA has a cross-attention transformer). It is hard to make conclusions based off limited approaches.

**Opportunities For Improvement:**

**Summary**

* More focus on the benchmark and its findings. There are only two models here and the paper focuses on potential fine-tuning but it does not talk about the original failures and weakness of each model.

* The clarity and motivation of the proposed approach Is important and needs improvement. The motivation for curating this dataset as opposed to using existing ones is also not entirely clear, could also be improved.

* More analysis of where these models are failing and why (this is difficult I know) they might be failing is also an area of improvement. It is claimed, although intuitive, that cross-attention will improve performance, but this is not justified using the proposed benchmark. Also, adding cross-attention adds complexity which is not discussed. This would make the paper and claims much stronger.

* Conclusions that drive future research should be clearer. One of the goals of benchmark studies is to find useful areas of improvement of future research. This is lacking or just not very clear.

**Details**

* There is nt overall insights to how models are behaving on the proposed dataset without finetuning, or they are very limited.

* Adding a cross-attention mechanism for CLIP’s visual and text representations is VERY similar to FLAVA’s architecture style. If this is not the case, then the method explanation is too brief and unclear. Also there are MANY approaches already proposing or using that. BridgeTower is a strong model in this and then there are even different architectures like ViLT that use a single-stream approach, maximizing the cross-attention. How would these approaches with cross-attention already heavily built in perform zero-shot on the proposed benchmark? This would be very interesting.

* Based on the language, it sounds like this work is just combining architectural designs from existing models after the CLIP model, which is not very novel. Furthermore, in the related works you talk about prompt-tuning but do not mention the additional models that you refer to under your methods description. If they are pivotal inspiration for the approach, they should briefly be explained so we can get better clarity on the proposed approach.

* The proposed approach adds a transformer but there is no discussion on the computational complexity. The reason prompt tuning was an interesting contribution was how little added complexity it had and how little it added to model capacity.

* Table 1 should have original model scores as well. Difficult to fully understand improvements otherwise.

* One issue is catastrophic forgetting. The approach should also be evaluated on other datasets like ImageNet to ensure the approach maintains existing “knowledge”. An example of what I mean by this is in Figure 3: https://openreview.net/pdf?id=KRLUvxh8uaX, "WHEN AND WHY VISION-LANGUAGE MODELS BE- HAVE LIKE BAGS-OF-WORDS, AND WHAT TO DO ABOUT IT?" published ICLR 2023.

**Relation To Prior Work:**

As for the modeling approach, these seem like extensions of existing works and not novel. There is not enough clarity on the benefits of this approach as it is not evaluated on the previous datasets CLIP is known for and there is no discussion on the computational benefits as well (prompt-tuning was proposed to reduce catastrophic forgetting and computational complexity/model capacity).

For the benchmark, there are several existing attribute-object datasets and they curate them together. It is hard to say if this dataset is more beneficial as there is little analysis/discussion on the benefits of this dataset and how it can be used to understand and benchmark weaknesses in existing approaches.

**Summary And Contributions:**

This work proposes a fine-tuning approach for CLIP that concatenates visual and text output and passes it to a transformer to apply self-attention. The [CLS] token that is pre-pended to the input to the self-attention module is used to compare to the original CLIP text features. By combining several datasets, they evaluate this approach on its ability to recognize attribute-object pairs and when there are multiple attribute-object pairs. The goal of this work is to: “adapt vision-language features to improve the compositional binding of attributes to objects” (see Supplementary L106). They evaluate the proposed approach on the proposed benchmark COLA that combines several attribute-object datasets from a variety of domains. They show their approach can improve performance compared to the original models and more traditional fine-tuning approaches.

---

> ### Author Response · Authors · 2023-08-16
>
> We thank you for the detailed and constructive comments. We will incorporate all the feedback in our paper.
>
> First, we would like to recap the contributions and goals of the paper.
>
> **What are we proposing?**  A new way to curate eval sets from existing datasets to test for multiple object-attribute compositionality. Using this benchmark, we analyze the design space of architectural choices and training paradigms commonly used that can best adapt VL models for this sort of compositionality.
>
> **What are we NOT proposing?** A new vision-language model or a new way to do vision-language fusion. We explore architectural choices commonly used in recently proposed VL models to see which one can adapt with the highest compositional performance on our benchmark.
>
> **What are the takeaways?**
>
> * Multimodal attention is better at distinguishing fine-grained attribute-object differences than split-modality attention.
> * Just using multimodal attention pre-training is not enough, we need to train them on attribute-object fine-grained data to perform well on such compositions.
>
> Now, we provide clarifications to the salient concerns raised:
>
> **The reviewer asked us to re-state our motivation for curating this dataset, given the existance of other compositional ones:**
>
> By “existing ones”, if the reviewer means existing object-attribute datasets, we build on top of existing datasets with attribute annotations (GQA, PACO, CLEVR, Visual Genome) to curate an eval set that specifically tests multiple attribute object compositionality. Other datasets that have attribute annotations involve single-attribute objects only on niche domains (like fashion, shoes, COCO-attributes etc).  By “existing”, if the reviewer means current compositionality benchmarks, the existing ones either do image-to-text (I2T) retrieval (CREPE, ARO), or test complex relationship compositions (Winoground). I2T is a simpler task than ours because text encoders are more robust than vision encoders at telling apart fine-grained differences, as shown by our experiments (simpler methods suffice on CREPE) and shown by Winoground as well. Why this is the case is still an open question. Further, Winoground is more complicated as it tests for more than just object-attribute compositions; however, it is significantly smaller in size. Additionally, we find that models fail to perform even on object-attribute compositions, demonstrating the possibility of using our datasets as a means to measure improvements in future models. Further, T2I attribute-object retrieval have higher practical relevance since in a collaborative setting, people will give instructions like “get the red wooden hammer”. These applications are outlined in lines 70-96 in the related work, but we will highlight them better in the paper.
>
> **The reviewer asked why COLA is harder:**
>
> This is still an open research question. We hope that our dataset will help drive answers to this question. We hypothesize that text encoders are more capable of understanding changes to the text input than image encoders are in representing similar changes in the image. This is why testing for compositionality using text-to-image (which requires distinguishing between difficult images to rank the correct ones higher) is so vital and is part of our contribution. We visualized some example queries where training on our data leads to the most improvements. These examples include images where attributes are attached to non-salient or smaller objects; e.g. a small blue circular sign beside a green bus. We also show we see minimal gains when the objects are large, such as blue sky, the original models are already good at this.
>
> **The reviewer wanted us to mention the failure cases of the original models:**
>
> We have included a few qualitative examples in the supplementary where we see the largest improvements from the original models and where we see minimal gains. We see large gains for attributes on smaller non-salient objects (Fig 4,5,6). We believe this is because of two reasons - 1) the COLA training data includes small objects with multiple attributes and requires the model to make this distinction. And 2)  the cross-attention between the image patches and the query token allow the model to learn these associations when being adapted more so than just tuning the CLIP original split-modality attention layers. This is a good suggestion and hence, we will move some of this analysis to the main paper.

---

> > ### Author Response · Authors · 2023-08-16
> >
> > **The reviewer asked for more analysis indicating when models fail:**
> >
> > We have included a few such examples in the supplementary where we show that queries related to size, closely related colors and spatial locations are especially difficult. We looked through more training examples and saw that this occurs because size is subjective, close colors are harder to disambiguate due to lighting differences (sometimes a light red brick wall almost looks similar to a white brick wall), and spatial relations annotations are noisy due to ambiguity in perspective (eg, left of something based on the viewer’s perspective or the image).  Although we do have these examples in the appendix, we will move a subset of them to the main paper and we will include this analysis.
> >
> > A systematic quantitative analysis of the types where we see larger and lesser gains beyond the qualitative examples requires more annotations to distinguish by types and to verify correct/noisy data annotations. While this is out of scope for this paper, we agree this is an important future direction.
> >
> > **The reviewer asked how COLA MAP differs from traditional MAP:**
> >
> > COLA MAP is a harder version of traditional MAP. COLA MAP is computed among hard distractors - this means we measure the precision of retrieving the images with the correct configuration of attributes and objects (eg, square white plate) from a set of images that also includes incorrect configurations of the same attributes and objects (eg square white table and a round plate). Other than that, it's the same MAP calculation. We explain COLA MAP in more detail in 126-130 in the main paper and 150-156 in the Supplementary. We see that COLA MAP is harder to improve on. _We can improve traditional MAP by more than 2x on GQA and 10x on CLEVR. In contrast, we only improve by about 1.09x on GQA and only about 2x on CLEVR on our harder COLA MAP._
> >
> > **The reviewer asked us to articulate the potential future research directions after our work:**
> >
> > We will make this clearer in the paper. From the quantitative results and qualitative analyses, these are the salient future directions we envision:
> >
> > * Training data that forces the model to reason about attributes and objects is highly crucial for the models to learn to attach the correct attributes to the correct objects. Hence, simple finetuning on our data improves performance. Hence, future research should focus on how we can create larger datasets with diverse attributes attached to different objects.
> > * For lightweight CLIP-style models, normal finetuning is not enough; we need to tune multimodal attention layers to allow the model to attach the right tokens in the query to the right image patches. However, simply adding a multimodal attention layer to pretrained models is not enough; they need to be adapted using attribute-object data to be able to reason about them. Hence, more research needs to be done on how we can effectively design multi-modal attention layers that induce compositional disentanglement between attributes and objects and other components of the scene.
> >
> > **The reviewer asked how the architectural designs we explore are related to FLAVA and other models:**
> >
> > Our multi-modal attention adapter is directly derived from FLAVA. Again, our contribution is not proposing new architectures. Our goal is to _analyze the design space of commonly used architectural choices_ to see which fusion method is best. Specifically, we explore existing architectures to study how they adapt models to attach the correct attributes to the objects compositionally. Hence, we test using FLAVA, ALBEF, MDETR, and FIBER as the choices (these attention choice results are in supplementary). Across all architectures, we see that _cross attention between image and text is the most salient feature in the attention design choice_ that leads to the most unseen performance gain for attribute-object compositions.

---

> > > ### Author Response · Authors · 2023-08-16
> > >
> > > **The reviewer was curious about how models that already have a cross-attention layer perform on the proposed benchmark when prompted zero-shot?**
> > >
> > > Great question. This is exactly the experiment we describe in lines 225, 226, and the results are shown in Table 1 and 2a. _Our adaptation on CLIP beats performance of using a model with cross-attention built-in zero-shot_. We compare our CLIP + MM-adapter to simply using FLAVA either zero-shot, or by fine-tuning the pre-trained cross-attention layers in FLAVA (FLAVA-FT late) in Table 1 and 2a. We see that our CLIP+MMAdapt outperforms even fine-tuning the cross-attention layers in FLAVA. The difference between our adapter and FLAVA is the training of the cross-attention modules from scratch on data with fine-grained differences in attributes and objects (square white plate vs round plate instead of random captions seen in FLAVA pretraining). Hence, it seems that having that kind of data is important for the cross-attention layers to robustly learn attribute-object compositions. We recognize that there are a flurry of other models with cross-attention as well by now, however, our adapter is the most similar in design to FLAVA and hence, we compare it with FLAVA.
> > >
> > > **The reviewer asked us to report the original model performance:**
> > >
> > > Original models underperform compared to our adaptations. We agree this is important to show as well, and hence, we will include these results in the table below. The Table below is the updated Table 1. The new rows we added CLIP-original and FLAVA-original based on your suggestion are italicized. Note that our MMAdapt on CLIP out-performs using FLAVA (with multimodal attention pre-trained) either original, linear-probed, or with fine-tuning their already pre-trained cross-modal attention layers (FT-Late). We will update this in the main paper; thanks for the suggestion.
> > >
> > > |                    |  GQA  |        |       | CLEVR |        |       |  PACO |        |       |
> > > |--------------------|:-----:|:------:|:-----:|:-----:|:------:|:-----:|:-----:|:------:|:-----:|
> > > |                    |  All  | Unseen |  Seen |  All  | Unseen |  Seen |  All  | Unseen |  Seen |
> > > | CLIP original      | _36.53_ |  _39.06_ | _34.24_ | _15.38_ |  _15.01_ | _15.32_ | _12.21_ |  _8.64_  | _15.79_ |
> > > | +Linear            | 40.44 |  42.87 | 38.24 | 47.96 |  29.43 | 46.75 | 14.22 |  6.75  | 21.68 |
> > > | + prompt-tune      | 37.40 |  40.69 | 34.43 | 29.61 |  23.17 | 28.05 | 12.76 |  5.92  | 19.61 |
> > > | + FT all           | 38.81 |  40.85 | 36.95 | 52.32 |  19.00 | 47.95 | 14.58 |  6.49  | 22.66 |
> > > | + FT late          | 42.19 |  44.61 | 40.01 | 64.06 |  27.53 | 67.48 | 15.66 |  8.74  | 22.58 |
> > > | + MM-Pred (our)    | 45.99 |  48.6  | 43.64 | 75.80 |  51.98 | 80.72 | 15.49 |  8.00  | 22.94 |
> > > | + MM-Adapter (our) | 46.83 |  48.86 | 44.99 | 88.21 |  89.52 | 77.00 | 18.56 |  11.47 | 25.66 |
> > > | FLAVA original     | _39.65_ |  _42.18_ | _37.37_ | _15.41_ |  _13.27_ | _15.93_ | _12.53_ |  _7.29_  | _17.76_ |
> > > | + Linear           | 37.07 |  39.96 | 34.46 | 19.30 |  17.53 | 18.52 | 11.65 |  7.90  | 15.39 |
> > > | + FT-late          | 39.58 |  42.26 | 37.16 | 77.95 |  72.72 | 66.42 | 12.82 |  5.79  | 19.84 |
> > >
> > > **The reviewer asked us if adaptation leads to catastrophic forgetting:**
> > >
> > > We would like to point out that our focus is not to propose a new foundation vision-language model that does well on all tasks. Our goal is to test common architectural choices for compositional performance by adapting them to the benchmark, where there are hard distractors of attribute-object combinations. Understandably, this adaptation results in some catastrophic forgetting. The ICLR paper mentioned does fine-tuning of CLIP, which we also compare to and show that a multimodal adaptation is better. We believe the findings of the ICLR paper are orthogonal - one can also do finetuning of our adapter using the hard negatives like they describe.
> > > However, to deliver on the reviewer’s request, **we ran an experiment to test our adapter on CIFAR**. Understandably, for zero-shot, we drop from CLIP’s original performance of 87% to 83%. However, we note that just by using 5% of the data of CIFAR (less than 1 epoch), **we reach 92% accuracy on CIFAR10 while maintaining performance on our COLA task**. This shows that the adaptation is effective for compositions (more so than fine-tuning) and can be made to remember other datasets too by larger scale training. We will add this result to the main paper and acknowledge a limitation that catastrophic forgetting while adapting is a major challenge that needs to be further investigated.

---

> > > > ### Author Response · Authors · 2023-08-16
> > > >
> > > > **The reviewer asked us to list the computational complexity of the adapters we add:**
> > > >
> > > > For the entire fine-tuning of CLIP we need to tune 151M parameters (FT all). In comparison, our MM-Adapt method only tunes 7.8M parameters. For fine-tuning only the late layers of CLIP (FT-Late), we tune 13M parameters comparable to our MM-Adapt. Linear probing is the only cheaper method to MM-Adapt (26K parameters). We discuss the number of parameters tuned for various fusion choices in the supplementary in Table 3. For FLAVA, tuning just the later cross-attention layers already inbuilt is 11M parameters. We will include this analysis in the main paper.
> > > >
> > > > **Minor clarifications:**
> > > >
> > > > **do you mean the CLS token is projected using a set of linear layers to make a final prediction?**
> > > >
> > > > Yes. We have a linear layer to map the 512-dim embedding to a binary class prediction.
> > > >
> > > > **Are you taking the [CLS] token from the self-attention between concatenated image and text features and comparing to the frozen text encoder whose output is also passed to the self-attention mechanism?**
> > > >
> > > > We are computing the cosine sim between [CLS] embedding and pooled text encoder embedding.
> > > >
> > > > **Is the only difference between MM-pred and MM-adaptor whether there is a set of linear layers after the self-attention for the [CLS] token only?**
> > > >
> > > > Yes. The only difference is linear layer vs a cosine similarity to the pooled text encoder output. We see this simple trick of computing cosine sim is like a regularizer and improves unseen compositions performance.
> > > >
> > > > **For the comment on more details on prompt tuning and how it differs from the CoOP[1] paper.**
> > > >
> > > > We apologize. Yes, we adopt a slightly modified approach from the [CoOP paper](https://arxiv.org/abs/2109.01134) [1]. Instead of tuning just the context, we tune the entire prompt - i.e the word embeddings for all the words in the prompt. This baseline is similar to [2]. We will make this clear in the paper.
> > > >
> > > > **Flava has a cross-attention module after already, are you adding an additional one?This is very unclear how it is applied to FLAVA. Are you removing its old one and just using the modality specific encoder outputs and using a different cross-attention module? Or you finetuning the cross-attention module FLAVA already has?**
> > > >
> > > > We try both. We tune the already available cross-attention (FLAVA+FT-Late in Table1 and 2a), and also remove those layers and add new ones from scratch (+MMPred, +MMAdapt). Interestingly, we see training the layers from scratch is much better than tuning them. Our hunch is because tuning probably adversarially perturbs the pre-trained local minima when taking gradient steps with low amounts of data. Further investigation on this is definitely required. We wrote this in results section line 263-263, 278-279. We will clarify this further to avoid confusion.
> > > >
> > > > **For the comment “another work that could be looked at is: [https://openreview.net/pdf?id=KRLUvxh8uaX](https://openreview.net/pdf?id=KRLUvxh8uaX) (published ICLR 2023)”**
> > > >
> > > > The ICLR paper is a great relevant related work, and we have included it in our related works section. The distinguishing factors are 1) we explore some architectural designs best for compositional adaptation, and 2) they use randomly shuffled patches of images to create hard image distractors. Instead, we use real images using scene graph annotations to mine for attributes and objects that are swapped.
> > > >
> > > > References:
> > > >
> > > > [1] Zhou, Kaiyang, et al. "Learning to prompt for vision-language models." International Journal of Computer Vision 130.9 (2022): 2337-2348.
> > > >
> > > > [2] Nayak, Nihal V., Peilin Yu, and Stephen Bach. "Learning to Compose Soft Prompts for Compositional Zero-Shot Learning." The Eleventh International Conference on Learning Representations. 2022.

---

> > > > ### Comment · Reviewer_H1og · 2023-08-21
> > > > **Include**
> > > >
> > > > Thank you for this detailed response! I think these are some important things that should be included in the final revision (if not already in the final revised version). I will increase my rating because of this.

---

> > > > > ### Author Response · Authors · 2023-08-31
> > > > >
> > > > > Thank you for your constructive review, and we are glad our reply clarified your doubts. We are incorporating all your feedback into our paper and will upload a revised PDF soon.

---

### Official Review · Reviewer_QhKB · 2023-07-20
**An interesting text-to-image retrieval benchmark for compositional groundings**

**Rating:** 7
**Confidence:** 4

**Strengths:**

1. The problem that this work is tackling is very interesting and practical, especially in the domain of grounded language learning and multimodal learning.
2. The gap in performance between the human baseline and SOTA large vision-language models indicates there is an ample opportunity for future research.
3. The method of selecting distractor images is efficient and makes the task more challenging which is evident based on the results when compared to another benchmark.

**Additional Feedback:**

None.

**Clarity:**

The paper is well-written and easy to follow. There are a few comments in the "Opportunities For Improvement" section.

**Correctness:**

The benchmark creation and evaluations seems to be valid, and the evaluations are complete and detailed.

**Documentation:**

The dataset is publicly available on GitHub, but the code is not available. The documentation could benefit from a script or an example code snippet showcasing how to use this benchmark.

**Ethics:**

In the supplementary material, the authors mention they obtained an exempted review from their IRB.

**Limitations:**

The authors talk about the limitations of their work and the fact that fine-tuning the pre-trained networks for this image retrieval task might interfere with the models' other capabilities such as question answering, and image captioning.

**Opportunities For Improvement:**

1. Figure 2 (b) could use more annotations or color-coding to match the caption. In general, the figure alongside the caption should be more self-sufficient to understand.

   a. Using different colors for frozen parts and trainable parts. Is blue for frozen parts and green for trainable parts? If yes, mention that in the caption or as a legend in the figure.

   b. Using same names in the caption and the figure, e.g. "Transformer Encoder" is the same as "multi-modal encoder" I believe.

   c. Does the $\otimes$ symbol represent cosine similarity? If yes, please specify.
2. Do all retrieval cases have two captions and two images? If yes, please specify that on line 48 where you say "... retrieve the correct image from a set of images." Also, are the number of captions and images different for single-object and multi-object cases? In Figure 3 (right), there are 3 images for one query. Please clarify this in the paper.
3. Continuing the last point, it seems to me that the number of image candidates for the retrieval task is different in single-object versus the multi-object case. Would you please clarify why did you make that design choice and did not use a consistent setup for both cases?
4. What is the reason that F1 score is not used instead of precision and accuracy?
5. On line 229, there is a missing word/number after the word Sec: "... as described in Sec using ..."
6. It would be more readable to have table 1 and table 2 (a) next to each other. Also, including the random performance for the single-object case would be informative and more consistent with the results for multi-object case.
7. In table 1, does row b (+prompt-tune) mean you add prompt-tune to CLIP+Linear or you use it instead of Linear resulting in CLIP+prompt-tune? Please clarify this either in caption or the entries of table.

**Relation To Prior Work:**

The authors compare their benchmark against another recent benchmark "CREPE" and position their finetuning methods very well with respect to related works.

**Summary And Contributions:**

The authors propose Cola, a text-to-image retrieval benchmark, to test the ability of large vision-language models when it comes to compositional groundings. In other words, this benchmark tests the models' ability to detect the correct image corresponding to the textual description of objects based on their attributes when there is a challenging alternative image (distractor image). Since the existing models struggle with this task, they also propose strategies to finetune the models on three datasets (GQA, CLEVR, and PACO).

This benchmark contains two types of queries: single-object and multi-object. Single-object queries describe one object only using its attributes and the other objects in the scene are not used in the description. The multi-object queries, on the other hand, use more than one object in the scene and their attributes in the description. The authors use the Visual Genome dataset to create the multi-object queries.
For each query, the authors use distractor images. In the single-object cases, they use images that contain at least one of the attributes, and in the multi-object cases, they use distractor image-query pairs where the attributes and objects are switched.
The test set is reviewed by humans to see if the paired queries and images match.

For finetuning, the authors use NCE Loss from contrastive learning to align queries and images. They first explore disjoint finetuning strategies such as linear-probing and prompt-tuning, both of which tune the parameters of the encoders for each modality separately. They then hypothesize that a cross-modal finetuning works better for the task of attribute-object binding, and they propose two strategies for joint finetuning of image and text: learning a linear classifier (MM-pred) to predict the output token and a multimodal-adapter (MM-adapter) to cross-attend to image and text representations resulting in a stronger representation that is later aligned with the frozen text embeddings.

To evaluate the performance for the single object queries, the authors use mean average retrieval precision (MAP), and for the multi-object queries, they use accuracy where the prediction is only correct if the prediction score of matching pairs of image and query is higher than the same query and the distractor image, for both queries.
The authors conclude that the existing large vision-language models (they use CLIP and FLAVA models in particular) perform poorly on their benchmark compared to another benchmark (suggesting this benchmark is more difficult) and compared to human baseline (suggesting that SOTA models are not good enough in grounding compositions).

**I would be happy to increase my rating** if the authors address the comments listed in section "Opportunities For Improvement."

---

> ### Author Response · Authors · 2023-08-15
>
> We thank you for the thoughtful and constructive reviews. We appreciate that the reviewer finds the problem “very exciting and practical”, that the “gap in performance between the human baseline and SOTA large vision-language models indicates there is an ample opportunity for future research” and that the “ method of selecting distractor images is efficient and makes the task more challenging” compared to other benchmarks.
>
> **[Question 1] The reviewer had suggestions for us to improve Figure 2:**
> We apologize for the confusion and will incorporate all the improvements. Yes, blue is frozen and green is trainable. We will add a legend to make this clearer. We will also make the naming consistent - transformer encoder and multi-modal encoder are the same.
>
> **[Question 2 and 3] The reviewer asked if all retrieval cases have two captions and two images or if the number of captions and images were different for single-object and multi-object splits? **
> Yes they are different. For the  single object split of COLA, we evaluate MAP on a set of 1950 query images. For the multi-object split, we evaluate using a two-way matching of two captions and two images- im1 should be matched to cap1 and im2 with cap2. We adopt this evaluation for the multi-objects split since there can be a combinatorially huge number of multi-object multi-attribute compositions, and it may be computationally expensive to measure MAP for all queries and for all images. On the other hand, computing a two-way accuracy is simpler and computationally cheaper. Further, a high accuracy on the two-way matching shows that the model is not confused by the attributes and objects in a different configuration and can identify the correct configuration of attributes and objects, which suffices for evaluating a model’s understanding of  attribute-object compositions. Finally, a contemporary paper on relationship compositionality, Winoground, also uses this method of evaluation, and hence, we keep the evaluation consistent to it.
>
> **[Question 4] The reviewer asked why we use precision and accuracy instead of F1:**
> Our evaluation measures are derived from prior literature. For multi-object split, we use accuracy to remain consistent with Winoground. For the single-object split, we agree F1 is also a valid metric as precision. Hence, we ran a sample of the evaluation using F1 on the GQA split of COLA single-objects and we see that all trends remain the same when comparing F1. For instance, F1 of CLIP baseline is 0.28, whereas FT all is 0.31, FT Late is 0.33, and our MM-Adapt is 0.39, MM-Pred is 0.40. Our conclusion stays the same: that adapting the multimodal attention layers is better than tuning the split-modal attention layers (FT-Late), fine-tuning the entire model, or linear probing. This is a good suggestion and we will include F1 scores to the Appendix.
>
> **[Question 5 and 6] The reviewer had some writing and organization suggestions.**
> We will incorporate your feedback.
>
> **[Question 7] The reviewer asked if by (+prompt-tune) in Table 1, we mean that we add prompt-tune to CLIP+Linear or use it instead of Linear:**
> We mean we use it instead of linear. So, we only prompt-tune CLIP. Each of the adaptations are independently applied to CLIP. We will make this clear in the paper.

---

> > ### Comment · Reviewer_QhKB · 2023-08-29
> > **Score updated, but looking forward for the revised pdf**
> >
> > Thank you for the explanations.
> > I thought the authors were going to upload a revised version of the paper with changes highlighted, but **I still don't see the changes in the paper**. I guess the authors meant that they would update the paper for the camera-ready version.
> > Anyway, **I increased my score based on the clarifications the authors provided.**

---

> > > ### Author Response · Authors · 2023-08-31
> > >
> > > Thank you for your thoughtful review! We are glad you liked the paper. We didn't realize we could edit the pdf during the review period. We apologize. We are adding all the changes now and will upload the revised PDF in a day or two.

---

### Official Review · Reviewer_dcEa · 2023-07-22
**Not a user of this kind of dataset but the details look reasonable to me**

**Rating:** 6
**Confidence:** 3
**Correctness:** Claims seem correct
**Clarity:** Yes

**Strengths:**

The paper is well organized and provides very clear examples of what is available in the dataset and how the dataset was collected. They also provide extensive benchmark using pre-trained and fine-tuned systems against their dataset

**Additional Feedback:**

N?A

**Documentation:**

A github page is provided but no licenses is mentioned / documentation on page could be improved to provide more details

**Ethics:**

I see no ethical concerns in this work

**Limitations:**

I see no particular limitations of this work (though again I don't work in VQA)

**Opportunities For Improvement:**

Some things I would have liked more detail on
- I would have liked to see some qualitative examples and commentary of the types of images that are hard in Cola (in comparison to CREPE)
- 10 crowd workers were used and you mention 83.88% human agreement on validation set, I would like to get more detail on what that means
- nit: Color scheme on table 2 seemed weird / wasn't immediately clear what color the reader should be paying attention to

**Relation To Prior Work:**

Yes

**Summary And Contributions:**

Cola paper provides a new text to image benchmark focused around object with attributes that are localized with respect to each other. I'm not in the VQA field so I don't know how significant / novel this work is, but the details provided look well reasoned about.

---

> ### Author Response · Authors · 2023-08-15
>
> Thank you for your positive and constructive review. We will incorporate your feedback in our paper and make the color scheme of Table 2 clearer.
>
> **The reviewer asked us to add more qualitative examples that are hard in Cola in comparison to CREPE:**
> This is a great suggestion. We find that CREPE is easier because the task requires matching a single query image to the correct text against a set of distractor texts. By contrast, COLA has two images and two texts, and hence, both texts must be matched to the correct images. From a qualitative analysis, we see that queries with closely related colors, sizes, and spatial relationships are difficult and found more often in COLA. Color is hard especially due to lighting differences. For instance, when a brick red wall looks close to white due to lighting, our model confuses it with a white wall. We have included some examples in the end of the supplementary, but we will add more such examples and some of this to the main paper as well.
>
> **The reviewer asked what we mean by 83.88% human agreement:**
> It means 8.33 out of the 10 workers on average agree on which image corresponds to which text in COLA. We will modify the sentence in the text to make this clearer.

---

### Official Review · Reviewer_pbZZ · 2023-07-26
**Review for the submission 299**

**Rating:** 7
**Confidence:** 4
**Correctness:** Yes, the experiments and evaluation a…
**Clarity:** The paper is well-written and organized.

**Strengths:**

The proposed benchmark is practical and could fill a gap in the task of compositional text-to-image retrieval. The two types of queries show a practical significance. Experiments seem to be sound in demonstrating the meaning of the proposed benchmark Cola.

**Additional Feedback:**

Please see above.

**Documentation:**

There is sufficient detail to support reproducibility.

**Limitations:**

The addressed limitations are worth exploring in the future work.

**Opportunities For Improvement:**

For the test set of single-object queries, the number of queries seems to be limited (320 for GQA, 96 for CLEVR, 400 for PACO). Whether using some larger datasets can lead to a larger number of test queries and thus allow for more comprehensive evaluation (especially for large models)?

**Relation To Prior Work:**

Discussion and comparisons with related benchmarks are appropriately made.

**Summary And Contributions:**

This paper studies the task of compositional text-image retrieval and proposes a new benchmark Cola, which focuses on the compositional attribute-object binding problems. The introduced problems are practical and interesting. It uses two vision-language models for evaluation. Extensive experiments with 6 fine-tuning strategies on these two models and comparison with the related benchmark CREPE show that Cola is a challenging task.

---

> ### Author Response · Authors · 2023-08-15
>
> Thank you for your positive and helpful review. We appreciate that the reviewer finds our “benchmark practical”, that it “could fill a gap in the task of compositional text-to-image retrieval”, and that our “experiments [are] sound in demonstrating the meaning of the proposed benchmark”. We will incorporate your feedback.
>
> **The reviewer asked whether using larger datasets can lead to a larger number of test queries and thus allow for more comprehensive evaluation:**
> The reviewer suggested a future direction of expansion to include a larger dataset with more test queries. Yes, we largely agree that a larger dataset with more test queries would allow an even more comprehensive evaluation. Since our task is to verify whether models can identify compositions of objects with multiple attributes against, we were limited by the availability of datasets with multiple (>1) attributes annotated per object. Although there are more classes available in datasets like PACO, increasing the number of classes in our benchmark increases the computation time required to evaluate all images for all compositional classes. Hence, we chose 200 randomly sampled seen and unseen classes each. However, we do agree having more classes will be beneficial for more extensive future analysis. We will include this statement in the paper.

---

### Decision · Program_Chairs · 2023-09-22

**Decision:**

Accept (Poster)

**Comment:**

The authors propose Cola, a text-to-image retrieval benchmark, to test the ability of large vision-language models when it comes to compositional groundings. The new dataset can help examine the models' ability to detect the correct image corresponding to the textual description of objects based on their attributes when there is a challenging alternative image (distractor image). All the reviewers appreciate the merits of the constructed Cola benchmark dataset, regarding that the task is interesting, challenge. Also the proposed approach shows noticeable improvements over the other fine-tuning approaches.

The paper can be accepted. The authors are highly suggested to consider the reviewers' comments and incorporate them into their final version and make corresponding revisions.